



# Evolution of Eddy Viscosity in the Wake of a Wind Turbine

Ryan Scott[1], Luis Martínez-Tossas[2], Nicholas Hamilton[2], and Raúl B. Cal[1]

[1]Portland State University Department of Mechanical Engineering, Portland, Oregon, USA
[2]National Renewable Energy Lab, Golden, Colorado, USA

**Correspondence:** Raúl Bayoán Cal (rcal@pdx.edu)

**Abstract.** The eddy viscosity hypothesis is a popular method in wind turbine wake modeling for estimating turbulent stresses. We document the downstream evolution of eddy viscosity in the wake of a wind turbine from experimental and large eddy simulation data. Wake eddy viscosity is isolated from its surroundings by subtracting the inflow profile and the driving forces are identified in each wake region. Eddy viscosity varies in response to changes in turbine geometry and nacelle misalignment with larger turbines generating stronger velocity gradients and shear stresses. We propose a model for eddy viscosity based on a Rayleigh distribution curve. Model parameters are obtained from scaling the eddy viscosity hypothesis and demonstrate satisfactory agreement with the reference data. The model is implemented in the curled wake formulation in the FLOw Redirection and Induction in Steady State (FLORIS) framework and assessed through comparisons to the previous formulation. Our approach produced more accurate flow field estimates with lower total error for the majority of cases.

## 1   Introduction

Accurate wake modeling is essential for optimizing wind plant layouts and creating effective control strategies (Veers et al., 2022; Meyers et al., 2022). Hybrid wake models balance the accuracy of high fidelity simulations with the computational efficiency of analytic models to facilitate wind plant design studies. While wake model development is an active area (Porté-Agel et al., 2020; Bastankhah et al., 2021, 2022), in the context of wind plant design, the applicability of these models is largely dependent on their ability to predict wake recovery on the order of turbine row spacing (Meyers et al., 2022). Subtle differences in estimating wake losses at this scale have an outsized impact on assessing the effectiveness of control strategies (Bay et al., 2022). Improving far wake representation without adding significant computational cost is needed to consider future wind plant design and operation strategies.

Hybrid wake models are often employed as design tools to evaluate the effectiveness of wake loss mitigation strategies such as wake steering, a popular approach for mitigating wake losses achieved by yawing or tilting the turbine rotor. While wake steering allows wind plant operators to increase net power production, the wake generated by a misaligned turbine introduces additional complexity requiring more advanced models (Martínez-Tossas et al., 2019; Zong and Porté-Agel, 2020). In particular, the formation of a counter-rotating vortex pair downstream of a misaligned turbine leads to substantial wake deformation and displacement (Howland et al., 2016; Bastankhah and Porté-Agel, 2016; Scott et al., 2020; Bossuyt et al., 2021). The curled wake formulation implemented in FLORIS (Flow Redirection and Induction in Steady State)(NREL, 2022a) was developed to model the effects of nacelle misalignment via yaw or tilt by using a collection of vortices shed from the





rotor plane (Martínez-Tossas et al., 2019, 2021). In order to maintain a balance of precision and efficiency, this model solves a simplified version of the Reynolds-Averaged Navier-Stokes equations with turbulence approximated by an effective eddy viscosity.

Eddy viscosity is responsible for relating the mean flow gradients to turbulent stress formation. In a wind turbine wake, eddy viscosity relates strain from momentum recovery to Reynolds stress formation. Ultimately, eddy viscosity in wake models determines the wake diffusion rate and is directly responsible for predicting wake longevity. Eddy viscosities are typically determined through a mixing length model and assumed to either maintain a constant value (Martínez-Tossas et al., 2019, 2021) or linearly increase with wake expansion (Shapiro et al., 2020; Bastankhah et al., 2022). Alternatively, constant eddy viscosities

can be modeled with a scalar function tuned to the turbulent production and dissipation of calibration flow (Van Der Laan et al., 2015). Eddy viscosities may also be obtained from measured or simulated flows via a linear regression between the strain rate tensor and turbulent shear stress tensor (Rockel et al., 2016; Bai et al., 2012). Rockel et al. (Rockel et al., 2016) found the eddy viscosity of a floating offshore turbine was affected by wave-induced pitch motion although current wake models do not include this information. Additionally the streamwise behavior of eddy viscosity has yet to be quantified in a parametric study

spanning multiple inflow conditions, turbine sizes, and misalignment angles. Finally, prior descriptions of eddy viscosity have relied on a single bulk value to represent turbulence in both the background and wake flows. This approach conflates boundary layer phenomena occurring at large scales with localized wake behavior.

    Here we propose a model to describe eddy viscosity as a function of downstream distance, inflow conditions, and turbine operating parameters. We isolate wake flow from its background by subtracting the inflow velocity profile. We document the

evolution of the eddy viscosity coefficient in the wake of a wind turbine for a range of conditions. Eddy viscosities are obtained from wind tunnel experiments with scaled turbine models and LES simulations performed in the Simulator for On/Offshore Wind Farm Applications SOWFA (Churchfield et al., 2012). Details regarding the theoretical background may be found in §2. Specifics on wind tunnel facilities, LES simulation procedures, and data processing are provided in §3. Findings and model development are presented in §4 with concluding remarks following in §5.

## 2   Theory

The Reynolds averaged Navier–Stokes equations are presented in tensor notation as:

$$u_j \frac{\partial u_i}{\partial x_j} = -\frac{1}{\rho} \frac{\partial p}{\partial x_i} - \frac{\partial \overline{u_i' u_j'}}{\partial x_j} - f_i \qquad (1)$$

where $\rho$ is the fluid density, $p$ the pressure, and $f_i$ is the force exerted by the turbine rotor. Here and in subsequent formulations, mean quantities are expressed as $(u)$ and turbulent fluctuations about the mean as primed $(u')$. Ensemble-averaging is denoted

with an overbar and subscript indices represent the streamwise $(u)$, vertical $(w)$, and spanwise $(v)$ velocity components in $x$, $z$, and $y$, respectively. Viscous terms are neglected as the wake flow is dominated by turbulent stresses and unsteady terms are omitted as the wake is considered stationary.



The eddy viscosity hypothesis relates the rate of strain tensor to the turbulent stress tensor allowing Eq. (1) to be described in terms of mean flow components only. This relationship is introduced as:

$$\overline{u_i'u_j'} = -2\nu_T S_{ij}, \tag{2}$$

where $\overline{u_i'u_j'}$ is the turbulent stress tensor and $S_{ij}$ is the rate of strain tensor. Eddy viscosity is written as $\nu_T$ and acts as a constant of proportionality between the turbulent stresses and rate of strain in eddy viscosity hypothesis. We propose $\overline{u_i'u_j'}$ and $\nu_T S_{ij}$ can be decomposed into background and wake flow components, denoted with subscripts B and w, respectively:

$$\overline{u_i'u_j'} = \overline{u_i'u_j'}|_{\text{B}} + \overline{u_i'u_j'}|_{\text{w}} \tag{3}$$

$$\nu_T S_{ij} = \nu_{T,\text{B}} S_{ij}|_{\text{B}} + \nu_{T,\text{w}} S_{ij}|_{\text{w}}, \tag{4}$$

where $\nu_{T,\text{B}}$ and $\nu_{T,\text{w}}$ are specific to the background and wake flows assuming both have analogous behavior such that independent values of $\nu_T$ can be assigned. Isolating the wake flow in this manner allows computing the wake contribution to Reynolds stresses as the difference between the flow upstream and downstream of the turbine. Since eddy viscosity relates the turbulent stress tensor to the rate of strain tenor, we can estimate this difference as the product of eddy viscosity and the wake rate of strain tensor. The resulting background and wake Reynolds stresses are:

$$\overline{u_i'u_j'}|_{\text{B}} = -2\nu_{T,\text{B}} S_{ij}|_{\text{B}} \tag{5}$$

$$\overline{u_i'u_j'}|_{\text{w}} = -2\nu_{T,\text{w}} S_{ij}|_{\text{w}}. \tag{6}$$

The total turbulent stress tensor, $\overline{u_i'u_j'}$, in the wake region can be reconstructed by adding the Reynolds stresses introduced by the wake Eq. (6) to the background flow Eq. (5) following Eq. (3) to produce:

$$\overline{u_i'u_j'} = -2\left[\nu_{T,\text{B}} S_{ij}|_{\text{B}} + \nu_{T,\text{w}} S_{ij}|_{\text{w}}\right]. \tag{7}$$

$S_{ij}|_{\text{B}}$ is assumed to maintain a constant value in a fully developed boundary layer implying $\nu_{T,\text{B}}$ is independent of $x$. Once the wake is fully recovered, the velocity deficit is no longer present i.e. $S_{ij}|_{\text{w}} = 0$ and the eddy viscosity hypothesis reduces to that of the background flow:

$$\overline{u_i'u_j'} = -2\left[\nu_{T,\text{B}} S_{ij}|_{\text{B}} + \nu_{T,\text{w}} \cdot 0\right] \qquad\qquad \overline{u_i'u_j'} = -2\nu_{T,\text{B}} S_{ij}|_{\text{B}} \tag{8}$$

Thus our efforts focus on modeling $\nu_{T,\text{w}}$ in the range where $S_{ij}|_w > 0$. Because we consider the background and wake flows separately, $\nu_{T,\text{w}}$ is determined from evaluating:

$$\overline{u_i'u_j'}|_x - \overline{u_i'u_j'}|_{\text{B}} = \nu_{T,\text{w}}\left[S_{ij}|_x - S_{ij}|_{\text{B}}\right], \tag{9}$$

where $x$ is given a distance downstream of the turbine. By performing this computation at multiple locations, we can detail the streamwise nature of eddy viscosity. A key assumption for assuming independent wake flow is neglecting ground interactions so the wake maintains a symmetric distribution of turbulent stresses at each downstream location. While this assumption holds for theoretical wakes, real turbines operate at a fixed distance above the ground over a variety of surfaces. The consequences of neglecting these ground interactions are detailed in the results and discussion.



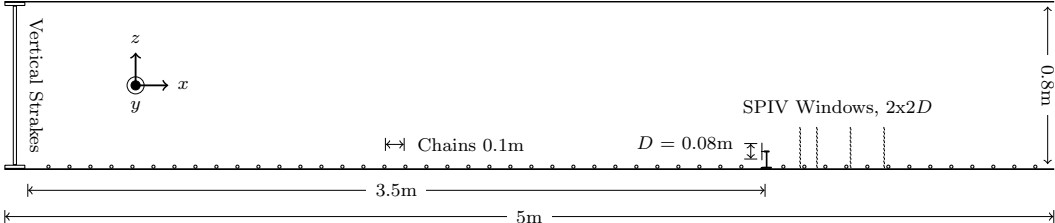

**Figure 1.** Portland State University wind tunnel with experimental apparatus and measurement locations to scale. Chains are shown at $2\times$ scale to enhance visibility. All dimensions in meters unless otherwise noted.

## 3 Methods

### 3.1 Experimental setup

Experimental data were collected through a series of wind tunnel experiments conducted by Bossuyt et al. (Bossuyt et al., 2021) in the Portland State University wind tunnel depicted in Fig. 1. The tunnel test section measures 5 m long with a cross section of $1.2 \times 0.8$ m. Inflow to the test section was conditioned with vertical strakes to produce a logarithmic boundary layer following Cal et al. and Hamilton et al. (Cal et al., 2010; Hamilton and Cal, 2015). Chains measuring 0.005 m were placed across the tunnel at fixed intervals of 0.1 m to maintain the boundary layer profile throughout the test section. A single scaled model turbine was positioned in the center of the tunnel 3.5 m downstream of the strakes. The model turbine measured 0.084 m in height with a rotor diameter of 0.08 m and was manufactured via 3D printing using a 3D Systems ProJet MJP 3600 printer. A Faulhaber 1016SR motor measuring 0.01 m in diameter was used as a DC generator. Turbine operation was controlled by means of a variable resistance potentiometer tuned such that the TSR measured $\lambda \approx 4$, $C_T \approx 0.65$, and $C_P \approx 0.15$ for a hub-height inflow velocity of $6.5$ ms$^{-1}$. The Reynolds number based on model turbine diameter was $3.3 \times 10^4$ for the chosen inflow velocity and the measured turbulence intensity was $11\%$. A reference case was created by orienting the model turbine normal to the inflow. Four yaw and tilt angles of $\pm 10°$ and $\pm 20°$ were considered by rotating the model turbine about its base. As a consequence, tilt misalignment varied nacelle elevation between 0.0045 m lower or higher and 0.026 m downstream or upstream, respectively. Stereoscopic particle image velocimetry (SPIV) measurements were recorded in the wake of a single model wind turbine at downstream distances of 2, 3, 5, and 7 rotor diameters. Inflow conditions were captured by removing the model turbine and recording free stream behavior. Neutrally buoyant aerosolized diethylhexyl sebacate seeding particles were maintained at constant density throughout the experiment. Measurement planes were oriented perpendicular to the mean flow and captured using two 4M pixel CCD cameras in conjunction with a Litron Nano double pulsed Nd:YAG (532 nm, 1200 mJ, 4 ns duration) laser. 1500 image pairs spanning $0.24 \times 0.18$ m were recorded at a rate of 4 Hz. Images were processed in LaVision DaVis 8.4 with decreasing multipass kernels of $48 \times 48$ px and $24 \times 24$ px at $50\%$ overlap for a spatial resolution of 0.001 m. Further details pertaining to the experimental setup including inflow profile measurements and turbine characterization are available in (Bossuyt et al., 2021).





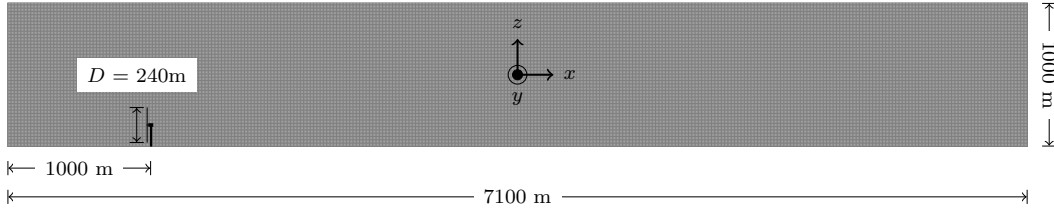

**Figure 2.** IEA-15 MW LES domain with turbine and grid spacing to scale. All dimensions in meters unless otherwise noted.

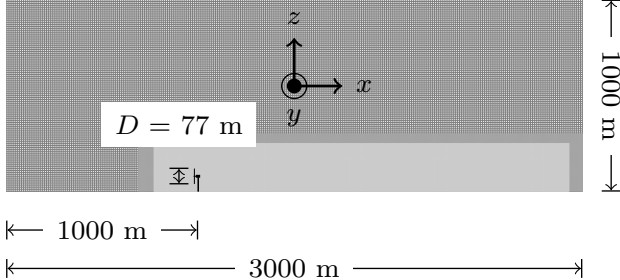

**Figure 3.** 1.5 MW LES domain with turbine, mesh refinement zones, and grid spacing to scale. All dimensions in meters unless otherwise noted.

## 3.2 LES setup

LES data were obtained from a series of SOWFA-6 (NREL, 2022c) simulations with the IEA-15 MW (Gaertner et al., 2020)

reference turbine and a 1.5 MW turbine shown in Fig. 2 and Fig. 3, respectively. The IEA-15 MW reference turbine rotor diameter measures 240 m and the hub height 150 m. The 1.5 MW turbine rotor diameter was 77 m and the hub height 80 m. Turbine behavior was simulated with the actuator disk model through OpenFAST coupling (NREL, 2022b). Yaw and tilt misalignments of $\pm 10°$ and $\pm 20°$ were imposed by rotating the turbine nacelle. The total domain size measured $(7,100; 3,000; 1,000)$ m for the 15 MW cases and $(3,000; 3,000; 1,000)$ m for the 1.5 MW cases with a grid resolution of 10 m. Two levels of mesh

refinement were added for the 1.5 MW cases to increase grid resolution in the turbine vicinity so the 15 MW and 1.5 MW cases shared similar grid resolution across the rotor. The first refinement zone spanned $(2,310; 390; 300)$ m and was located at $(690; 1,320; 0)$ m. The secondary refinement measured $(2,160; 310; 250)$ m and was located at $(770; 1,350; 0)$ m. Grid resolution was increased to 5 m in the first zone and again to 2.5 m in the second. A single turbine was located at $(1,000; 1500; H)$ m for all SOWFA-6 simulations. A neutral atmospheric boundary layer inflow was generated with a 20,000 second precursor

simulation on each base domain with a hub-height inflow velocity of 8 ms$^{-1}$. In each simulation, 500 seconds were allotted for startup transients followed by 3,000 seconds of data collection.



## 4 Results and discussion

### 4.1 Streamwise evolution of eddy viscosity

Eddy viscosity values are obtained at each downstream location from the slope of a least squares linear regression between
$S_{13}|_w$ and $\overline{u'w'}|_w$. The streamwise-vertical components of the Reynolds stress and rate of strain tensors are selected as they
are of the greatest magnitude in the wake and are responsible for the majority of energy flux into a wind plant (Porté-Agel
et al., 2020; Scott et al., 2020). Wake flow is isolated downstream of the turbine following Eq. (3) and Eq. (4). Both $S_{13}|_B$ and
$\overline{u'w'}|_B$ are obtained from the reference plane at $X$ for experimental data then as streamwise averages spanning $x = -1.25D$
through $x = -0.25D$ for simulation data. In the experimental setup, $\partial W/\partial x = 0$ upstream of the turbine by design and is
not included in $S_{13}|_B$. Furthermore, $\partial W/\partial x$ is found to be at least two orders of magnitude less than $\partial U/\partial z$ across all data
sets. However, its contribution to $S_{13}|_w$ is included for completeness. Example planes of the streamwise-vertical component
of the velocity gradient tensor and its corresponding component of the Reynolds stress tensor are shown in Fig. 4. Slope
fit error is computed with the standard regression error and presented as shaded $95\%$ confidence bounds on $\nu_{T,w}$ in Fig.
4. Since ground interactions introduce substantial strain near the the surface, data below a height of $z = H - D/1.75$ are
discarded to ensure eddy viscosities reflect wake rather than surface phenomena. Additionally, only data contained within
$(y \pm 1.15D, z \pm H - D/1.75)$ are considered at each streamwise location so the measurement area is consistent across all data
sets. Neglecting LES data outside this area does not influence the quality of the linear regression as quantities outside the wake
are near zero from Eq. (9).

Eddy viscosity evolves downstream of the turbine as the wake flow recovers, highlighted in Fig. 5. Immediately downstream
of the turbine rotor, $x/D \lesssim 1$, the flow is dominated by the momentum deficit and pressure gradients from turbine operation.
Shear stresses are small in this region and develop downstream as energy is converted from mean flow gradients into turbulence.
A linear relationship is present between $S_{13,w}$ and $\overline{u'w'}|_w$ with negligible average fit errors below $1\%$ of $\nu_{T,w}^\star$. $S_{13}|_w$ peaks just
behind the rotor and begins to decay as the initial momentum deficit recovers. Eddy viscosity displays a convex increase in
response to the decreasing rate of strain coupled with increasing turbulent stresses.

From $1 \lesssim x/D \lesssim 3$, the momentum deficit recovers and strain transfers to turbulence with maximum shear stresses present
near $x/D \approx 3$. Linear regression error is low here as well with average fit errors near $2\%$ of $\nu_{T,w}^\star$. $S_{13}|_w$ diminishes from
continued wake recovery while Reynolds stresses continue to develop. However, after $\overline{u'w'}|_w$ peaks near $x/D = 3$, wake eddy
viscosity is driven by the rate of momentum recovery relative to turbulent dissipation.

Beyond, the remaining velocity deficit is recovered and turbulence follows the energy cascade towards small scales. Since
$S_{13}|_w$ and $\overline{u'w'}|_w$ are decreasing, a shift from concave to convex curvature is visible in Fig. 5. As the wake dissipates, the
quality of the linear fit decreases as shown by increasing regression error magnitudes in Fig. 5 up to $10\%$ of $\nu_{T,w}^\star$. The rate of
strain is small, $S_{13,w} \approx 1 \times 10^{-4}$, but remains as long as turbulent fluctuations are present in the wake. Eddy viscosity reaches
its maximum here as the mechanisms for continued growth have deteriorated and $\nu_{T,w}$ is now driven by exclusively Reynolds
stress decay. At extreme downstream distances, $x/D \gtrsim 20$, the eddy viscosity hypothesis no longer holds as both $S_{13}|_w$ and

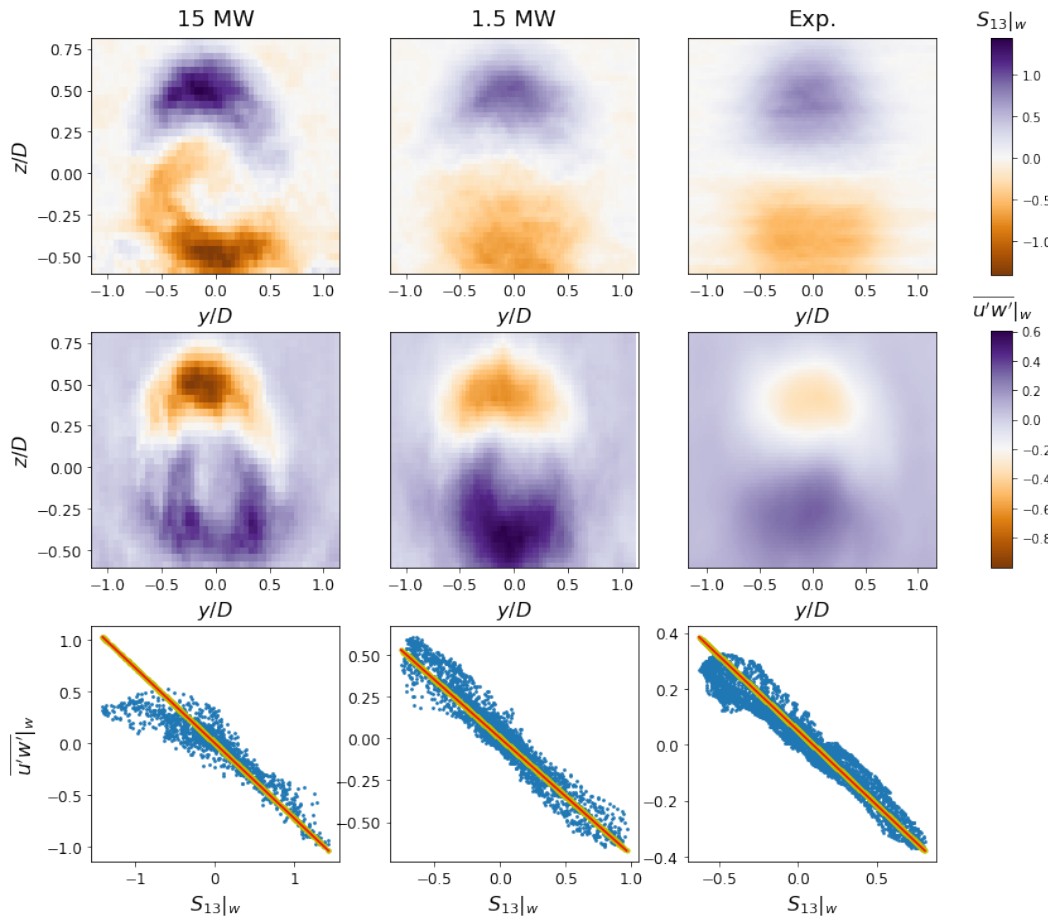

**Figure 4.** Contours of streamwise velocity in $y - z$ planes of $S_{13}|_w$ (top) and $\overline{u'_i u'_j}|_w$ (middle) with the corresponding linear fit (bottom) at $x/D = 3$ for the aligned turbine cases. The 1.5 MW (center) and scaled model turbine (right) display symmetry in both $S_{13}|_w$ and $\overline{u'_i u'_j}|_w$ as the wakes from these turbines are unimpeded. The 15 MW turbine (left) produces a greater velocity deficit than the smaller turbines and exhibits additional strain near the ground due to its low rotor-ground clearance.

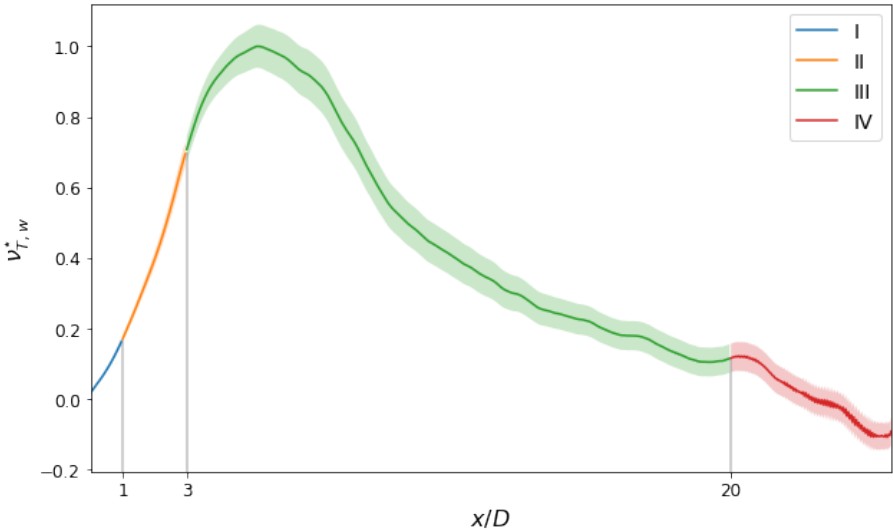

**Figure 5.** Normalized wake eddy viscosity for the aligned 15 MW LES case. Slope fit error is represented with shading. Color denotes portions of the wake governed by similar terms in Eq. (1). Downstream locations where driving terms transition are indicated with vertical lines.

$\overline{u'w'}|_\text{w}$ are near zero. The linear regression is highly sensitive to noise and produces erroneous values as the wake has returned to the background flow.

Based on these observations, we propose the streamwise behavior of eddy viscosity in the wake may be modeled as a Rayleigh function:

$$\nu_{T,\text{w}}(x) = A\frac{x}{\sigma^2}e^{-x^2/2\sigma^2}, \tag{10}$$

where $A$ parameterizes amplitude and $\sigma$ is the scale parameter. $A$ is determined by scaling the eddy viscosity hypothesis with a velocity scale, $U_s$, a length scale, $l_s$, and noting $\partial W/\partial x << 1$:

$$U_s^2 \sim 2\nu_{T,\text{w}}\left[\frac{U_s}{l_s}\right], \tag{11}$$

Rearranging Eq. (11) to isolate eddy viscosity yields:

$$\frac{l_s U_s}{2} \sim \nu_{T,\text{w}} \tag{12}$$

For the wake velocity scale $U_s$, $U_s \sim U_\text{B}\sqrt{1-C_T}$ is chosen following Bastankhah et al. (2016) and for the length scale $l_s \sim R$. Substituting the velocity and length scales into Eq. (12):

$$\nu_{T,\text{w}} \sim \frac{RU_\text{B}\sqrt{1-C_T}}{2} \tag{13}$$

where $C_T$ is the turbine thrust coefficient, $U_\text{B}$ is the inflow velocity at hub height, and $R$ is the rotor radius. Radius is selected rather than diameter as both the rate of strain tensor and shear stresses are symmetric about the wake center. The scale parameter

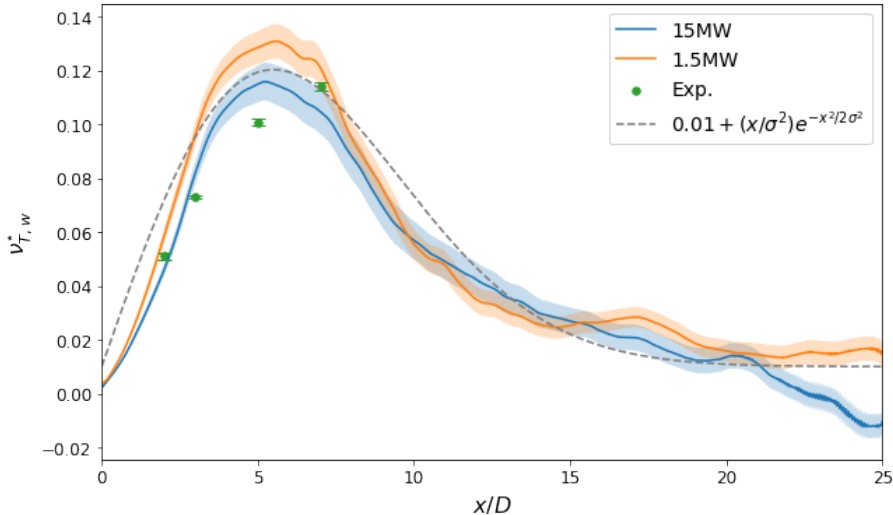

**Figure 6.** Scaled streamwise wake eddy viscosity for aligned turbine cases with fit error and the proposed model Eq. (14). Note experimental measurements only extend to $7D$ downstream so far wake recovery is exclusively from simulated data. Curves are depicted for SOWFA-6 LES cases as the streamwise resolution is sufficiently high.

$\sigma$ is obtained by substituting $A$ and fitting Eq. (10) to the eddy viscosity curves for aligned turbines. The parameter $\sigma$ represents a characteristic distance of where the wakes are in a fully developed far-wake state. Substituting for $A$ and $\sigma$ into Eq. (10) produces:

$$\nu_{T,\mathrm{w}}(x) = A \left[ 0.01 + \frac{x}{\sigma^2} e^{-x^2/2\sigma^2} \right] \tag{14}$$

where $A = R U_{\mathrm{B}} \sqrt{1 - C_T}/2$ and $\sigma = 5.5$. A constant offset of $0.01$ is added to prevent $\nu_{T,\mathrm{w}} = 0$ immediately behind the rotor
and far downstream as this would imply wake diffusion is absent. This scaling nondimensionalizes eddy viscosity from each data set and demonstrates agreement across the range of turbine sizes in Fig. 6.

### 4.2    Eddy viscosity of misaligned turbines

The impact of nacelle misalignment on eddy viscosity is presented in Fig. 7. Despite the presence of the counter-rotating vortex pair, the streamwise evolution of eddy viscosity is consistent with earlier trends. However, introducing wake deflection reduces
the peak viscosity magnitude as the momentum deficit from a misaligned turbine is lower than that of a turbine operating in nominal conditions. Additionally, the formation of the counter-rotating vortex pair downstream of a misaligned turbine serves to deform and deflect the wake accelerating its recovery and lowering $\nu_T(x)$. This is particularly the case for $20°$ tilt with the 15 MW reference turbine since the large deflection angle and turbine aspect ratio combine to drive the wake into the ground where it experiences rapid dissipation.

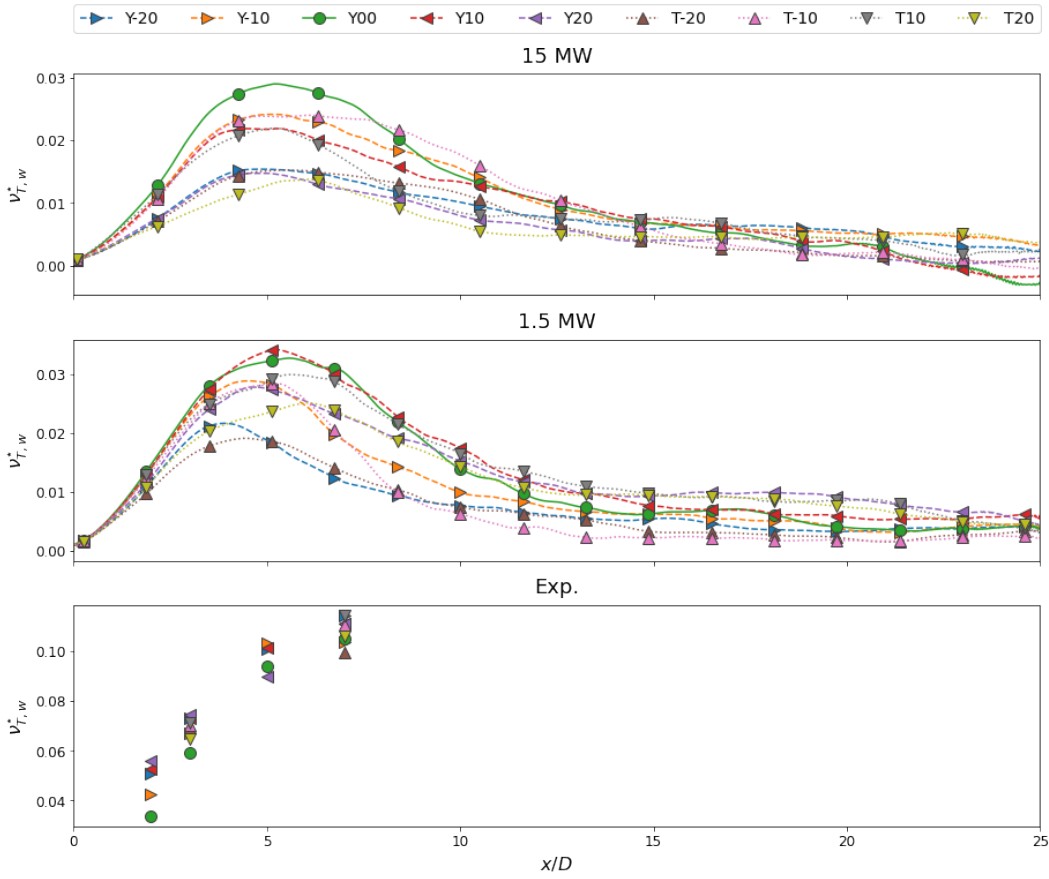

**Figure 7.** Scaled streamwise wake eddy viscosities for 15 MW (top), 1.5 MW (middle), and scaled model turbine (bottom) cases. Nacelle misalignment reduces peak eddy viscosity by modifying wake recovery. Large misalignment angles produce weak wakes with strong counter-rotating vortex pairs and low peak eddy viscosities. Note experimental data is only available to $7D$ downstream and yawed cases contain two measurements at $x/D = 3$ and $x/D = 7$.

In the proposed formulation, nacelle misalignment is accounted for by varying the thrust coefficient. Yet this parameter alone is insufficient to capture the dynamics of the curled wake as evidenced by the departure of the eddy viscosity curves for yawed and tilted turbines from the unmodified case. While the normal velocity imparted by the counter-rotating vortex pair increases $\partial w/\partial x$, this term remains at least two orders of magnitude below $\partial u/\partial z$ and as such has a negligible effect on eddy viscosity. The primary drivers are variations in Reynolds shear stress and surface interactions with the ground resulting from wake deflection. Under yaw misalignment, the wake experiences asymmetric growth as momentum is entrained into the wake center by the counter-rotating vortex pair (Howland et al., 2016; Bastankhah and Porté-Agel, 2016; Scott et al., 2020; Bossuyt et al., 2021). Asymmetric expansion forces the lower portion of the wake close to the ground where it experiences additional strain leading to heightened Reynolds shear stresses. Ground interactions are inherent to tilt misalignment as this approach





either by directs the wake up into the boundary layer where it is advected by the mean flow or into the ground (Fleming et al.,
2014; Annoni et al., 2017; Scott et al., 2020; Bossuyt et al., 2021). The extent of additional strain from surface interactions
depends on the wake expansion rate, nacelle deflection angle, height of the bottom tip above the ground, and local surface
characteristics. Describing this interaction is beyond the scope of the present work although such a study is well warranted.

### 4.3 Model implementation

The proposed model Eq. (14) is incorporated within the FLORIS curled wake formulation to assess the effectiveness of includ-
ing a high fidelity eddy viscosity representation in wake modeling applications. For the experimental comparison, the boundary
layer height in FLORIS was reduced to match the wind tunnel layer height described by Bossuyt et al. (Bossuyt et al., 2021).
Streamwise flow fields were computed for each turbine type, misalignment angle, and inflow velocity. Contours of streamwise
velocity in $y - z$ planes of each case are presented at $x/d = 3$ in Fig. 8 (yaw) and Fig. 9 (tilt). Quantitative comparisons are
computed as error between flow field estimates and the corresponding experimental or simulation data with:

$$\epsilon(x) = 100\% * \frac{\ell^2(U_x - U_{x,F})}{\ell^2(U_x)} \tag{15}$$

where $\ell^2$ is the Euclidean norm, $U_x$ is the measured or simulated streamwise flow field at a given downstream location $x$, and
$U_{x,F}$ is the flow field estimate at $x$. Error using the proposed model is presented for each case in Fig. 10.

Maximum error occurs immediately downstream of the turbine as FLORIS is not designed to represent near wake phenom-
ena. Non-zero error is expected from comparing a wake modeling utility with experimental or LES data. Additionally, because
the curled wake model uses a mirror condition at the ground to satisfy no-slip, estimated flow fields have a steeper boundary
layer profile than the reference data. Aligned turbine cases produce the greatest errors since the proposed model over-predicts
near wake diffusion. Similarly, low misalignment angles are similar to typical turbine operation and the proposed model un-
derestimates the velocity deficit for these cases as well. Overall, flow field error decreases with distance behind the turbine
and reaches a minimum by $x/D = 5$. However, beyond $x/D = 15$ error increases since the proposed model limits turbulent
stress formation and thus wake diffusion. Error increase the fastest under positive tilt deflection since wakes in this scenario
are deflected into the ground.

### 4.4 Model comparison

Comparisons between the proposed model and existing formulation are considered to evaluate the impact of increasing eddy
viscosity fidelity in the curled wake model. Contours of streamwise velocity in $y - z$ planes are presented in Figs. 13, 14, and
15. Flow field error is computed for the current formulation following Eq. (15) and shown in Fig. 11. Relative error between
flow field estimates is presented in Fig. 12 and calculated with:

$$\Delta\epsilon(x) = 100\% * \frac{\ell^2(U_x - U_{x,\nu_T=C}) - \ell^2(U_x - U_{x,\nu_T=f(x)})}{\ell^2(U_x - U_{x,\nu_T=C})} \tag{16}$$



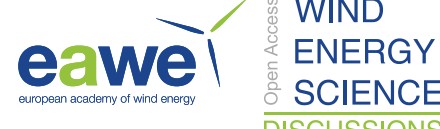

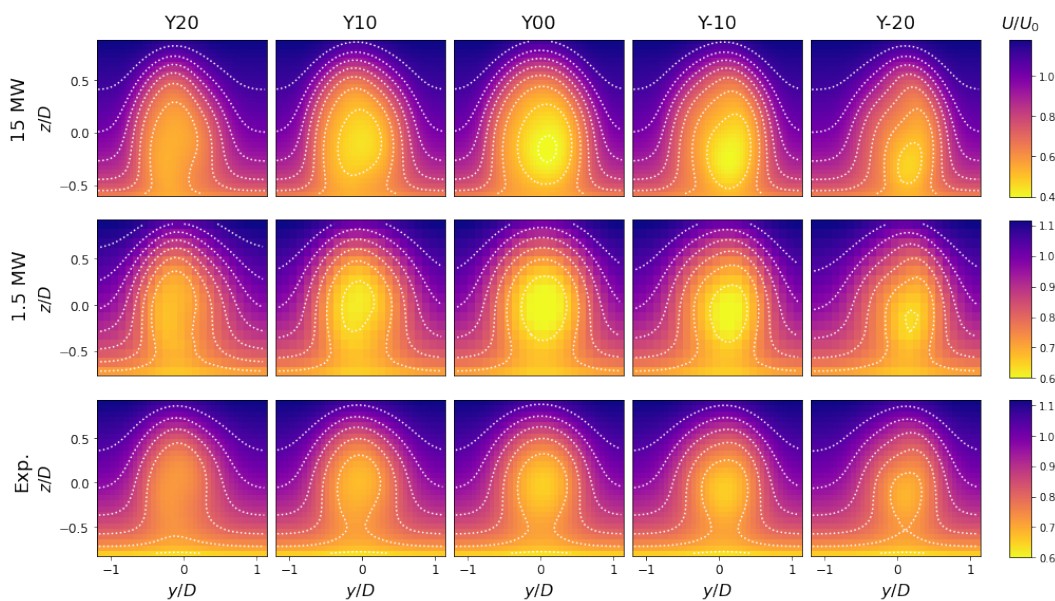

**Figure 8.** FLORIS streamwise flow fields across yaw angles at $x/D = 3$ for 15 MW (top), 1.5 MW (middle), and scaled model turbine (bottom) inputs. Wake deformation is visible for yawed cases since the curled wake model includes the counter-rotating vortex pair.

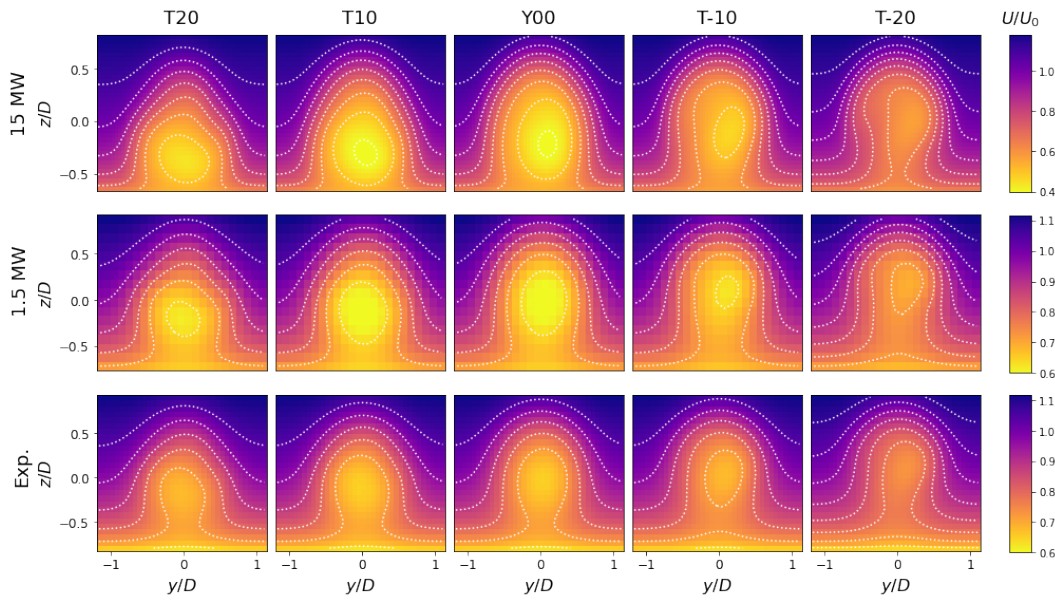

**Figure 9.** FLORIS streamwise flow fields across tilt angles at $x/D = 3$ for 15 MW (top), 1.5 MW (middle), and scaled model turbine (bottom) inputs. The curled wake model captures the effects of tilt mislaignment with asymmetric wake deformation observed across turbine sizes.





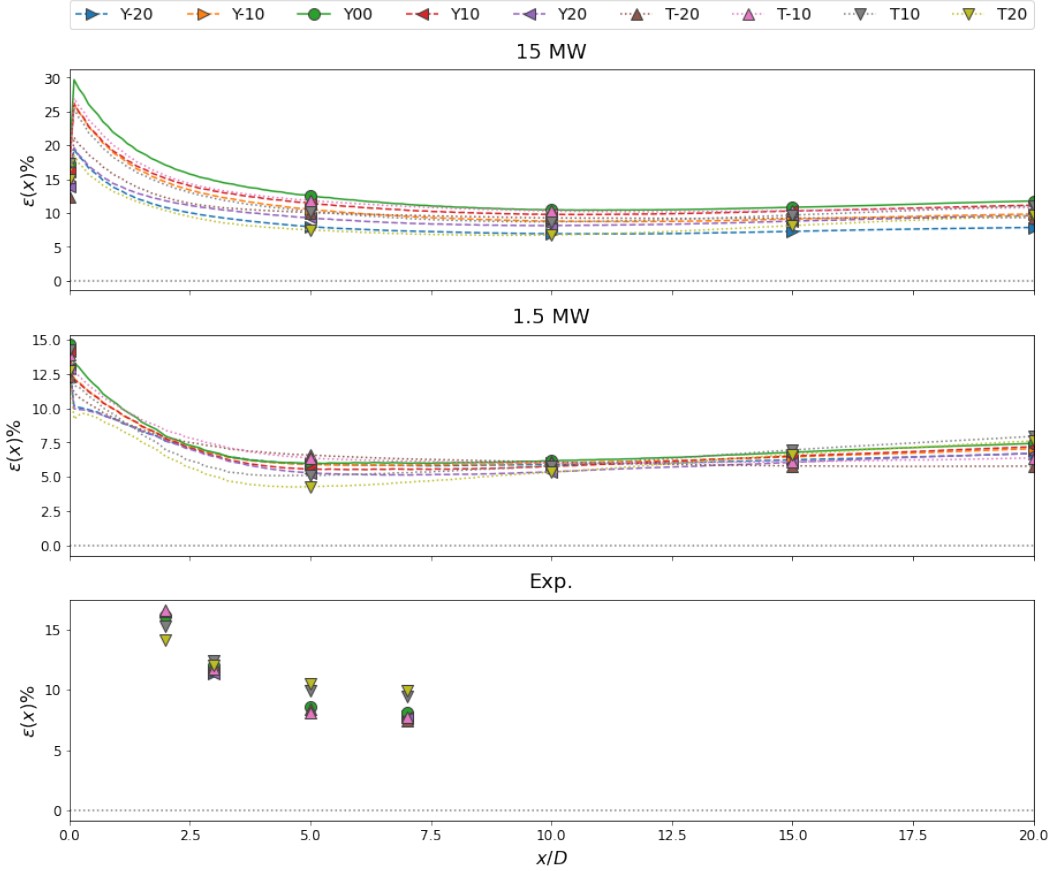

**Figure 10.** FLORIS streamwise flow field error from the proposed model for 15 MW (top), 1.5 MW (middle), and scaled model turbine (bottom) inputs. Error is greatest in the near wake as FLORIS is not designed to represent near-wake phenomena. Flow field error decreases downstream of the turbine reaching a minimum near $x/D = 5$. In the far wake, error increases from over-predicting the wake velocity deficit.

where $\ell^2$ is the Euclidean norm, $U_x$ is the measured or simulated streamwise flow field at a given downstream location $x$, $U_{x,\nu_T=C}$ is the flow field estimate computed with the existing formulation, and $U_{x,\nu_T=f(x)}$ is the flow field estimate computed with the proposed model.

Error peaks immediately downstream of the turbine with a secondary peak visible near $x/D = 2$ for the 1.5 MW cases. As noted previously, such errors are expected as FLORIS is not designed to model near wake phenomena. The presence of a secondary peak for the 1.5 MW turbine implies the assigned eddy viscosity value and thus near wake diffusion is too high. There is not a clear trend between nacelle misalignment and error with the current approach. Because the existing formulation maintains a constant eddy viscosity, error is relative to how well the prescribed value represents wake diffusion for a given turbine. For the 15 MW cases, the aligned turbine produces the greatest error while for the 1.5 MW cases, positive tilt deflection





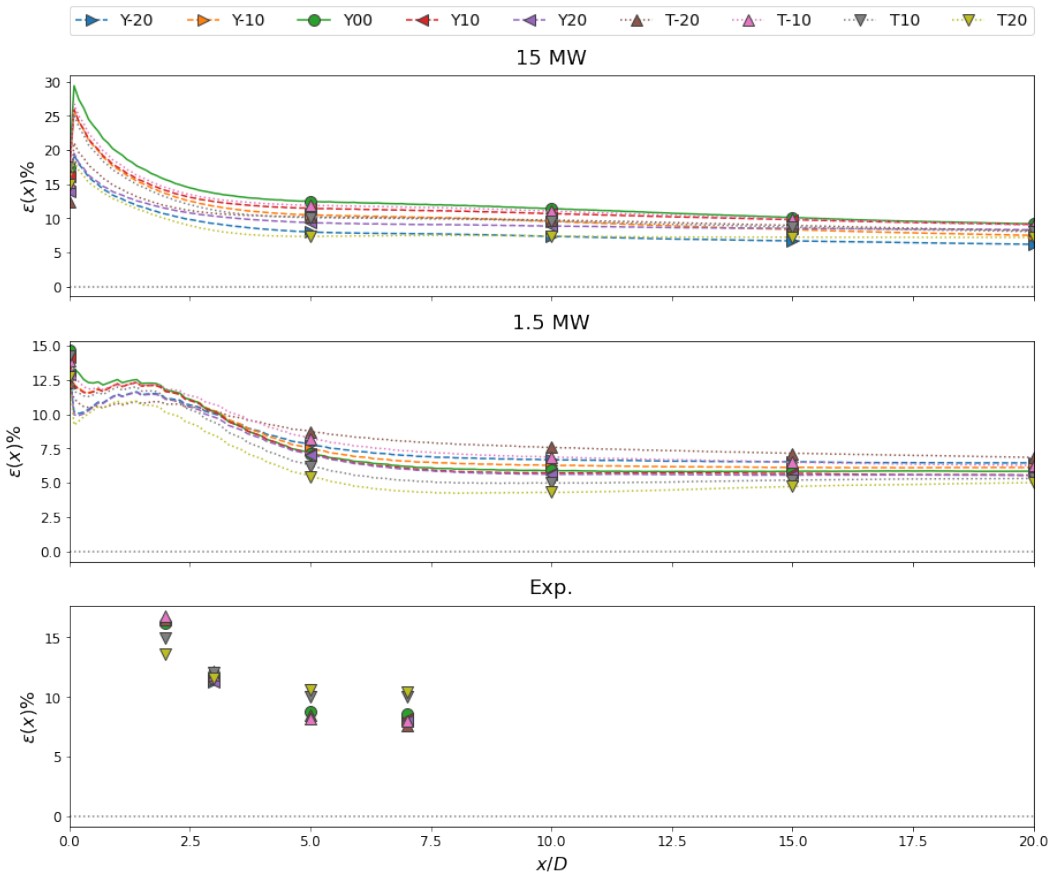

**Figure 11.** FLORIS streamwise flow field error using the existing formulation for the 15 MW (top), 1.5 MW (middle), and scaled model turbine (bottom) inputs. Maximum error is present directly behind the turbine with a secondary peak near $x/D = 2$ for the 1.5 MW cases. Error decreases with distance downstream for all cases except positive tilt angles which attain a local minima near $x/D = 7.5$.

results in greater error. Flow field error decreases with distance behind the turbine for all cases except positive tilt deflection which results in increased wake diffusion by directing the wake into the ground.

Qualitatively, the proposed model captures the wake shape and velocity deficit as shown in Figs. 13, 14, and 15. However,
both formulations over-predict the far wake velocity deficit for the 15 MW turbine with a lingering momentum deficit visible in Fig. 13. Despite its large velocity deficit, the 15 MW turbine has low ground clearance such that the wake experiences shear from the ground and dissipates within a relatively short distance. Consequently, the total error for the 15 MW cases is large compared to the relative error between eddy viscosity formulations. Further downstream, the impact of the curled wake mirror condition is apparent with a region of low velocity extending up from the ground into the wake flow. Since the boundary layer
profile is assigned by the curled wake model, this region is present in all flow field estimates.





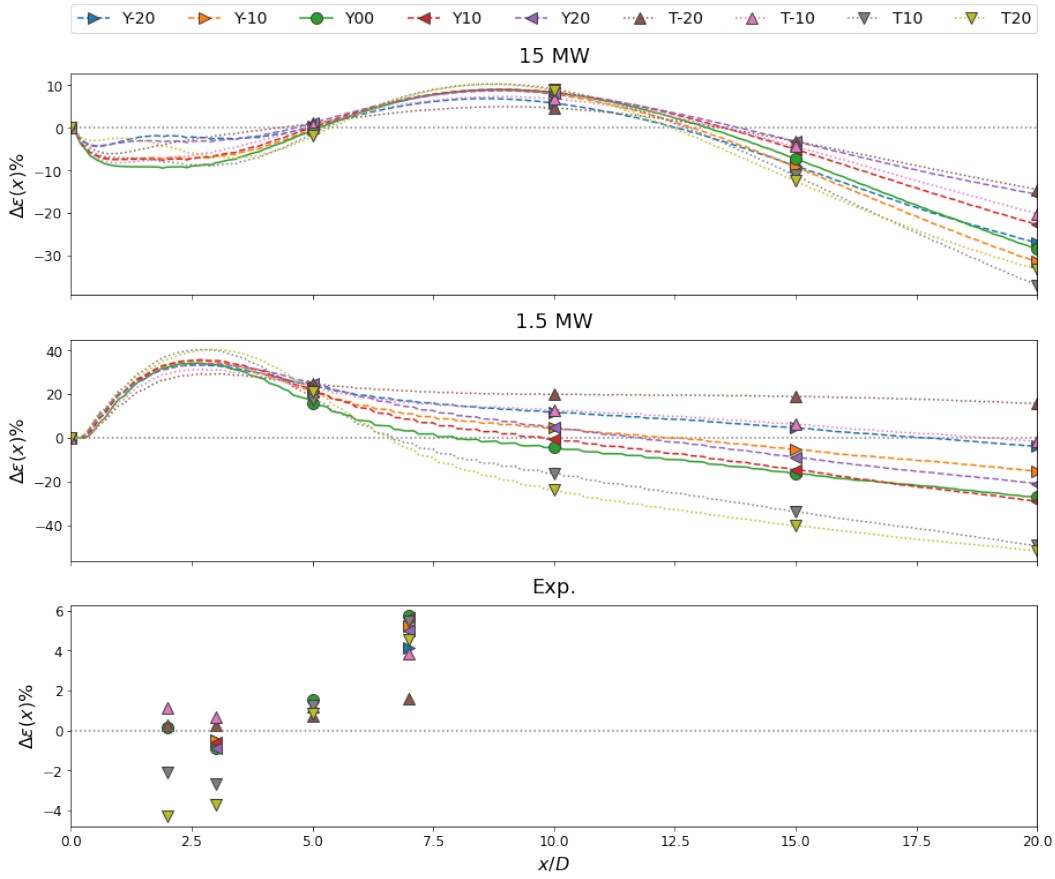

**Figure 12.** Relative error between eddy viscosity formulations for the 15 MW (top), 1.5 MW (middle), and scaled model turbine (bottom) cases. Positive values indicate regions where the proposed model reduces flow field error relative to the proposed formulation while negative values show areas where the proposed model introduces additional error.

The proposed model outperforms the existing formulation for the majority of cases. Although relative error is inconsistent in the near wake, our model reduces net error for $x/D \leq 15$. Because the proposed model limits Reynolds stress production, and thus wake diffusion, it is less accurate far downstream of the turbine due to over-predicting the wake velocity deficit. While nacelle misalignment is not accounted for in either formulation outside of the thrust coefficient, the proposed model produces
less error for negative tilt misalignment at all downstream locations. Under these circumstances, the wake is deflected into the boundary layer and advected downstream rather than deformed. As such, it maintains a coherent structure and lingering velocity deficit which the proposed model reproduces. The improvement in flow field prediction for these cases is attributed to better representation of Reynolds stress formation and is not assumed to convey additional fidelity in modelling the impact of the counter-rotating vortex pair on wake recovery. The notable exception is for positive tilt angles where the wake is directed

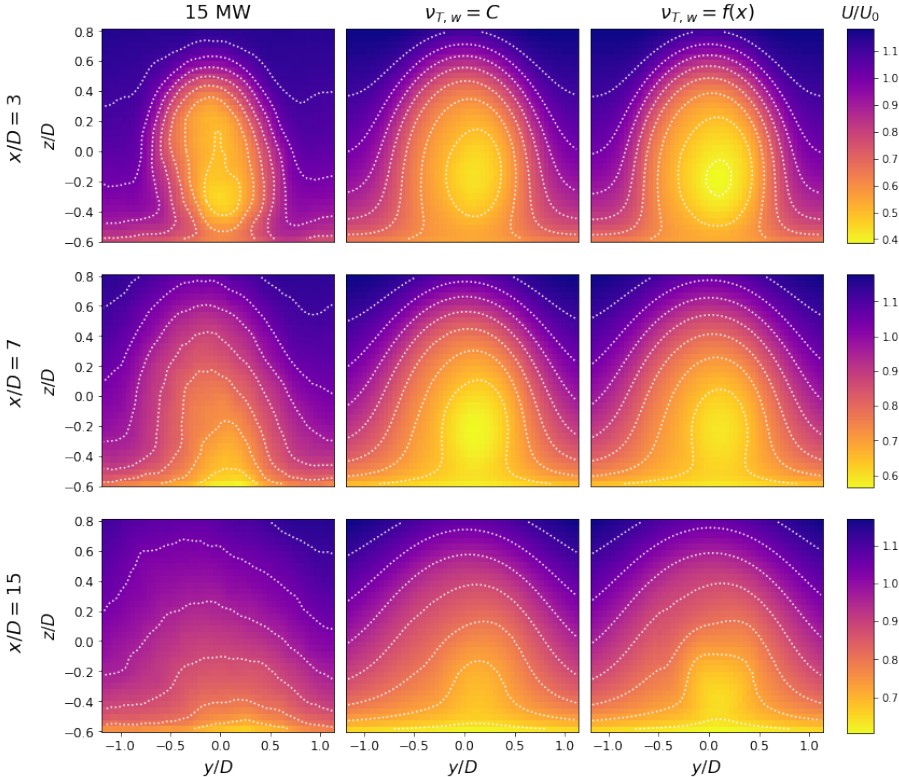

**Figure 13.** Contours of streamwise velocity in $y - z$ planes for the aligned 15 MW turbine. Both formulations capture the near wake profile but over-predict the far wake velocity deficit.

into the ground. As discussed previously, the proposed model over-predicts the wake velocity deficit leading to comparatively large errors.



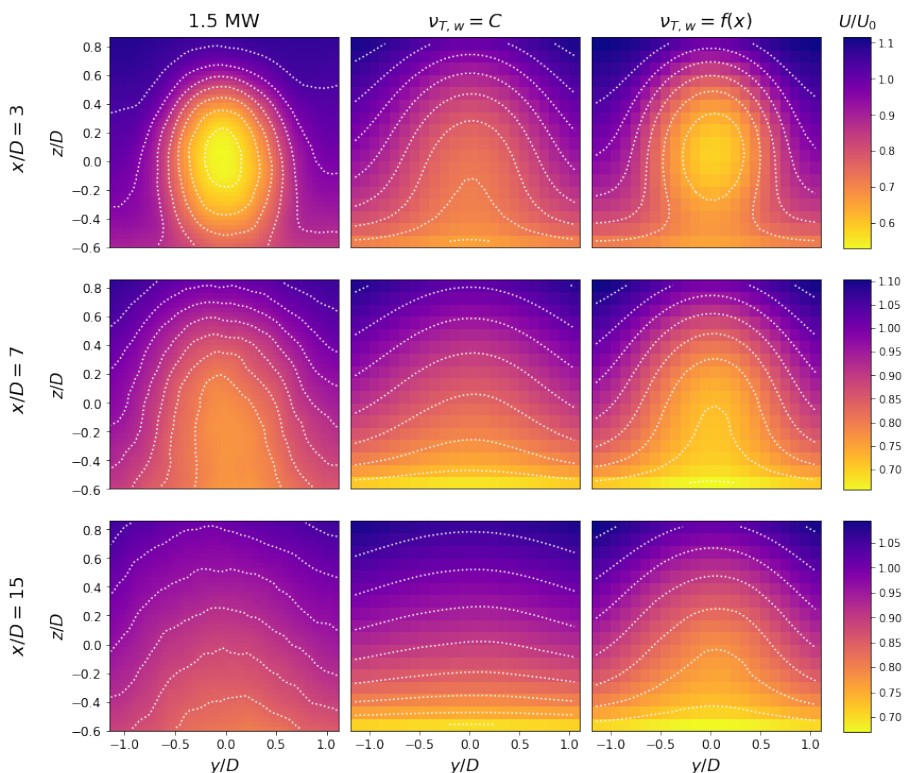

**Figure 14.** Contours of streamwise velocity in $y - z$ planes for the aligned 1.5 MW turbine. The proposed model replicates the near wake velocity deficit as well as the far wake profile while the current formulation under-predicts at all downstream locations. The velocity gradient introduced by the mirror condition in the curled wake model is visible at $x/D = 7$ and $x/D = 15$.

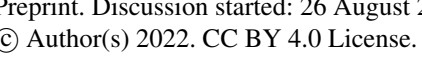

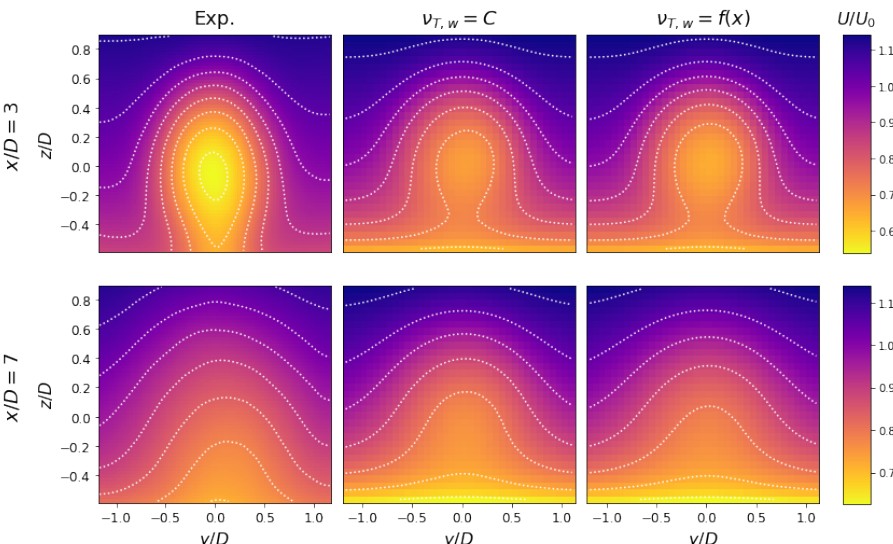

**Figure 15.** Contours of streamwise velocity in $y - z$ planes for the aligned scaled model turbine. Here both formulations miss the near wake velocity deficit but recreate the velocity profile at $x/D = 7$. Despite scaling the boundary layer height in FLORIS, the gradient produced by the mirror condition in the curled wake model is visible at $x/D = 7$.





# 5 Conclusions

We presented the streamwise evolution of eddy viscosity in the wake of a wind turbine for various turbine types and inflow conditions. Eddy viscosities were obtained from a linear correlation between the rate of strain and turbulent stress tensors in the wake. Wake flow was isolated from the background by subtracting the inflow profile. Eddy viscosities were then nondimensionalized through scale analysis which produced satisfactory agreement between data sets.

In the near wake, eddy viscosity depends on momentum recovery and is governed by the rate of strain tensor. Immediately behind the rotor, the velocity deficit is at its maximum and recovers in the near wake as momentum is entrained from the surrounding flow. The strain induced by this near wake momentum recovery is responsible for the initial increase in eddy viscosity. Turbulent shear stresses, however, take time to form which provides eddy viscosity with consistent growth as turbulence is produced in the wake. Here eddy viscosity is driven by the interplay between the remaining rate of strain tensor and Reynolds shear stress formation. In the far wake, eddy viscosity mimics turbulent decay as the wake dissipates. Reduced eddy viscosities were observed for deflected wakes as misaligned turbines extract less power from the inflow. The formation of the counter-rotating vortex pair was linked to asymmetric wake expansion which in turn produced complex surface interactions between the wake flow and ground. Wake deformation from the counter-rotating vortex pair and the ground invalidate the requirement of a symmetric turbulent stress distribution needed to apply the eddy viscosity hypothesis. As a consequence, further work is needed to characterize the far wake recovery of misaligned turbines.

A model for the streamwise evolution of eddy viscosity was proposed based on a Rayleigh probability density function. The model was incorporated into the FLORIS curled wake model and compared to the existing eddy viscosity formulation. The proposed model outperformed the existing approach with a net improvement in estimating experimental and LES flow fields. The model performed best for cases where the wake maintained a coherent structure and velocity deficit far downstream. However, our model under-estimated the initial velocity deficit and far wake recovery. Because the model limits eddy viscosity in the far wake, turbulent stress formation and thus diffusion are constrained leading to higher velocity deficits in the far wake. While an empirical modification to the underlying Rayleigh function is possible, a more robust description of far wake recovery is needed to address this shortcoming.

Our model improves upon current formulations by capturing the streamwise evolution of eddy viscosity. This approach reduces net error in flow field estimation when incorporated into the FLORIS curled wake model. Representing the exchange between rate of strain and Reynolds stresses increases wake modeling fidelity and will allow hybrid wake modeling utilities to better-predict wake recovery in wind plant settings. Additionally, future work can resolve the discrepancies reported for nacelle misalignment. Describing surface interactions in terms of turbine operating parameters and roughness height is one promising avenue for further refinement. We anticipate future developments in this area will lead to improved predictions of wind plant performance and enable the design of more efficient wind plants.

*Author contributions.* All authors contributed equally to each aspect of this work.





*Competing interests.* Raúl Bayoán Cal serves as an associate editor, the remaining authors declare no competing interests.

290   *Acknowledgements.* This work was authored in part by the National Renewable Energy Laboratory, operated by Alliance for Sustainable Energy, LLC, for the U.S. Department of Energy (DOE) under Contract No. DE-AC36-08GO28308. Funding provided by the Department of Energy Office of Energy Efficiency and Renewable Energy Wind Energy Technologies Office. The views expressed in the article do not necessarily represent the views of the DOE or the U.S. Government. The U.S. Government retains and the publisher, by accepting the article for publication, acknowledges that the U.S. Government retains a nonexclusive, paid-up, irrevocable, worldwide license to publish or

295   reproduce the published form of this work, or allow others to do so, for U.S. Government purposes.

    A portion of The research was performed using computational resources sponsored by the Department of Energy's Office of Energy Efficiency and Renewable Energy and located at the National Renewable Energy Laboratory.





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
