# Peer review of "Evolution of Eddy Viscosity in the Wake of a Wind Turbine"

_Wind Energy Science, 2022_

## Referee Comment (RC1)

**Review of *Evolution of Eddy Viscosity in the Wake of a Wind Turbine* by Ryan Scott et al.**

Reviewer: M. Paul van der Laan, DTU Wind Energy

November 7, 2022

The authors propose an analytical expression for the streamwise evolution of the eddy viscosity in a wind turbine wake using large eddy simulation (LES) cases and wind tunnel measurements. The proposed model is implemented in an engineering wake model developed for deflected wakes and its performance is tested against LES.

The article is relatively well written and include novel ideas. I enjoyed reading about the analytic form of the eddy viscosity of a single wake; I think this is useful to the wind energy community because it could be employed in other engineering wake models as well.

There are a number of major issues with the equations and assumptions, which are listed below; they need to be addressed before the article can be considered for publication in Wind Energy Science.

**Main comments**

1. There are a number of methods to determine an eddy viscosity from (LES) data. For example, one could also use the direct definition of the eddy viscosity from a two equation model (e.g. $k$-$\varepsilon$: $\nu_T = C_\mu \frac{k^2}{\varepsilon}$), or a more complex relation following higher order turbulence models, see for example [1]. Here, the difficulty is to determine $\varepsilon$, which some authors obtain from solving an $\varepsilon$ transport equation on the reference data (so as a post processing step), see for example [2]. I think it make sense to add such a discussion to the introduction.

2. Your chosen method for obtaining an eddy viscosity from a data set is not well motivated and tested. If you use the Boussinesq approximation, then there are 5 equations but you only have one unknown: the scalar eddy viscosity. Hence, the system is over determined. You have chosen to only look at $\overline{u'w'}$ and then neglect the other equations and you motivate this in Lines 129-132: *Eddy viscosity values are obtained at each downstream location from the slope of a least squares linear regression between $S_{13}|_w$ and $\overline{u'w'}|_w$. The streamwise-vertical components of the Reynolds stress and rate of strain tensors are selected as they are of the greatest magnitude in the wake and are responsible for the majority of energy flux into a wind plant (Porté-Agel et al., 2020; Scott et al., 2020).* I agree that the shear-stresses are the main contributors to the wake recovery, however, $S_{13}|_w$ does not have to the largest component, it can also be the lateral Reynolds-stress $\overline{u'v'}|_w$, depending on the inter wind turbine spacing in the wind farm, see for example [3]. In a recent work of my own [4], I showed/visualized that the peaks of the shear stresses indicate how much wake recovery takes places in the lateral and vertical directions. You could look at your own LES and wind tunnel data to see which is dominant one. Alternatively, you could also write the Boussinesq hypothesis in polar coordinates and then you could choose to only use the equation for the shear stress in the radial direction for calculating the eddy viscosity from the data. This also makes sense for your application where the employed engineering wake models uses axi-symmetry. If you choose to stick with the current method, you should at least show how large the errors are from the unused equations of the Boussinesq hypothesis, especially the one for the lateral shear stress.

3. If you want make general conclusions about the proposed analytical model of the eddy viscosity then I would recommend to investigate aligned single wake wind turbines first using a range of thrust coefficients and turbulence intensities, since these two parameters are the main ones. I am aware that misaligned turbines will typically have lower thrust coefficients, but you will also have a lateral thrust component that changes the wake shape. (Atmospheric stability is also important and could be added in future work.)

4. What is a Hybrid wake model? In the introduction you write that a Hybrid wake model has the computational efficiency of an analytic/engineering wake model. To me that sounds like a Hybrid wake model falls into the class of engineering wake models. In my personal opinion, any model that does not solve a momentum equation iteratively (i.e. including pressure gradients) is a low fidelity / engineering wake model.

5. Lines 34-36: You write *Alternatively, constant eddy viscosities can be modeled with a scalar function tuned to the turbulent production and dissipation of calibration flow (Van Der Laan et al., 2015).* Note that we are not modeling a constant eddy-viscosity in the work that you refer to. We actually use a three-dimensional scalar eddy viscosity similar to the standard $k$-$\varepsilon$ model. The model is then improved by multiplying the eddy viscosity with a scalar function that is dependent on the local flow. Effectively, this scalar function limits the eddy viscosity in regions where the flow is far from its equilibrium, i.e. the near wake. The model can also be interpreted a turbulence length scale limiter, see for example [5]. In theory, you could use the $k$-$\varepsilon$-$f_P$ definition to determine the eddy viscosity, which should lead to better prediction of $\nu_T$ in the near wake, but it could be future work to test other forms of $\nu_T$. (A small note: my is spelled as *van der Laan*.)

6. Equations (2), (5)-(9) are not complete and should include a term with the turbulent kinetic energy, $k$:

$$\overline{u_i'u_j'} = \frac{2}{3}k\delta_{ij} - 2\nu_T S_{ij} \tag{1}$$

If you aim to neglect $k$ you should mention this and motivate reader that this makes sense. Later on you calculate the eddy-viscosity from only $\overline{u'w'}$, so I understand where omitting $k$ comes from. You could move the motivation from that part (Section 4.1) to theoretical part (Section 2). In addition, I can think equations (7) and (8) can be removed, because the previous equations already define everything.

7. Line 77: You mention that $S_{ij}|_B$ is assumed to be constant in the fully developed ABL. However, even for a simple neutral surface layer following a logarithmic law we get that $S_{ij}|_B$ is far from constant and scales with $1/z$: $S_{ij}|_B = 1/2u_*/(\kappa z)$. If you aim to only look at one height, e.g. hub height, then I can follow your assumption but this does seem to be case in the remaining part of the article (i.e. Figure 4).

8. Line 81: You mention *Thus our efforts focus on modeling $\nu_{T,w}$ in the range where $S_{ij}|_w > 0$.* However, $S_{ij}|_w > 0$ can also be negative in the wake. (That is why $\overline{u'v'}$ has both positive and negative values in the wake.)

9. Section 3.2, LES setup. It is not clear to me what kind of neutral ABL is simulated. Is this a pressure-driven ABL or is this an ABL including an inversion and Coriolis forces? The first type of ABL should be Reynolds number independent for a fixed ratio of the domain height $L_z$ and roughness length, $z_0$, i.e. $z_0/L_z$ is constant. In that case, the two LES cases mainly differ in wind turbine model if you have scaled the ABL inflow such $z_H/L_z$ is constant. I expect the thrust coefficient to be main difference between the two LES cases, it would be helpful to report the mean thrust coefficient value for both of them. The thrust force distribution could also play a role in the near wake but is not expected to have a large impact on the far wake; a similar argument can be applied for the ground clearance, $z_H/D$ and other design parameters.

10. Line 160: you mention *At extreme downstream distances, $x/D$, the eddy viscosity hypothesis no longer holds as both $S_{13}|_w$ and $\overline{u'w'}|_w$ are near zero.* I do not agree, the eddy viscosity hypothesis holds better for a flow that is in near equilibrium (the far-far wake or the background flow) with respect to the (near) wake. I guess your main point refers to the regression method not being able to calculate an eddy-viscosity with the proposed method. Hence this part needs some rewriting.

11. Equation (11) and (12): I do not understand where Equation (11) comes from, it is not the Boussinesq hypothesis (without $k$) because you lack the gradients and the minus sign. In addition, you could arrive at Equation (12) in a more simply way by dimensional analysis since we have that the unit of $\nu_T$ is m$^2$/s thus one can write $\nu_T = l_s U_s$. In addition, your chosen $U_s$ is based on the far wake of 1D momentum theory, which I think you should mention.

12. Lines 258-259: You conclude to have investigated multiple inflow conditions, but you mainly used one LES ABL inflow as the wind tunnel wake data was not sufficient to capture the entire wake.

13. Conflict of interest: I guess this can be removed as Jens Sørensen is the editor. It should not be a problem if a coauthor is also an editor in the journal, as long as he or she is not the handling editor.

14. I think you author contribution statement is too vague. Please clarify who did what.

**Minor comments**

1. You sometimes write *LES simulation*. This means that you write the word *simulation* twice so I would suggest to remove the second word.

2. Equation (1): You forgot to mention that you assume an incompressible flow.

**References**

[1] M. Baungaard, S. Wallin, M. P. van der Laan, and M. Kelly, "Wind turbine wake simulation with explicit algebraic reynolds stress modeling," *Wind Energy Science*, vol. 7, no. 5, pp. 1975–2002, 2022. [Online]. Available: https://wes.copernicus.org/articles/7/1975/2022/

[2] M. Schmelzer, R. P. Dwight, and P. Cinnella, "Discovery of algebraic reynolds-stress models using sparse symbolic regression," *Flow, Turbulence and Combustion*, vol. 104, no. 2, pp. 579–603, 2020. [Online]. Available: https://doi.org/10.1007/s10494-019-00089-x

[3] J. Meyers and C. Meneveau, "Flow visualization using momentum and energy transport tubes and applications to turbulent flow in wind farms," *Journal of Fluid Mechanics*, vol. 715, p. 335–358, 2013. [Online]. Available: https://doi.org/10.1017/jfm.2012.523

[4] M. P. van der Laan, M. Baungaard, and M. Kelly, "Brief communication: A clarification of wake recovery mechanisms," *Wind Energy Science Discussions*, vol. 2022, pp. 1–8, 2022. [Online]. Available: https://wes.copernicus.org/preprints/wes-2022-56/

[5] M. P. van der Laan and S. J. Andersen, "The turbulence scales of a wind turbine wake: A revisit of extended k-epsilon models," *Journal of Physics: Conference Series*, vol. 1037, no. 072001, p. 1, 2018. [Online]. Available: https://doi.org/10.1088/1742-6596/1037/7/072001

---

## Author Comment (AC1)

The authors propose an analytical expression for the streamwise evolution of the eddy viscosity in a wind turbine wake using large eddy simulation (LES) cases and wind tunnel measurements. The proposed model is implemented in an engineering wake model developed for deflected wakes and its performance is tested against LES.

The article is relatively well written and include novel ideas. I enjoyed reading about the analytic form of the eddy viscosity of a single wake; I think this is useful to the wind energy community because it could be employed in other engineering wake models as well.

There are a number of major issues with the equations and assumptions, which are listed below; they need to be addressed before the article can be considered for publication in Wind Energy Science.

**Response:** The authors appreciate this review and the constructive comments. The text has been edited to reflect the reviewers comments and for clarity. The reviewers comments are addressed below and further revisions are highlighted in **bold** in the text.

**Comment 1:** There are a number of methods to determine an eddy viscosity from (LES) data. For example, one could also use the direct definition of the eddy viscosity from a two equation model (e.g. $k - \varepsilon$: $\nu_T = C_\mu \frac{k^2}{\varepsilon}$), or a more complex relation following higher order turbulence models, see $\varepsilon$ for example []. Here, the difficulty is to determine $\varepsilon$, which some authors obtain from solving an $\varepsilon$ transport equation on the reference data (so as a post processing step), see for example []. I think it makes sense to add such a discussion to the introduction.

**Response:** Thank you, we have added the suggested discussion to the introduction and revised the surrounding text. The updated paragraph now reads:

"Eddy viscosity is responsible for relating the mean flow gradients and turbulent kinetic energy to turbulent stress formation. In a wind turbine wake, eddy viscosity relates strain from momentum recovery to Reynolds stress formation. Ultimately, eddy viscosity in wake models determines the wake diffusion rate and is directly responsible for predicting wake longevity. Eddy viscosities are typically determined through a mixing length model and assumed to either maintain a constant value [9, 10, 7] or linearly increase with wake expansion [16, 2]. Alternatively, eddy viscosity can be modeled with a scalar function tuned to the turbulent production and dissipation of calibration flow [6, 8]. If high resolution data are available, such as from large eddy simulations (LES) or Reynold-averaged Navier-Stokes (RANS) models, eddy viscosities may be obtained by directly solving the Boussinesq approximation or higher order closure models [14, 4]. Eddy viscosities may also be obtained from measured or simulated flows via a linear regression between the strain rate tensor and Reynolds shear stress tensor [13, 1]. Across techniques, prior descriptions of eddy viscosity have relied on a bulk value to represent turbulence in both the background and wake flows. This approach conflates boundary layer phenomena occurring at large scales with localized wake behavior. Additionally, Rockel et al. [13] found the eddy viscosity of a floating offshore turbine was affected by wave-induced pitch motion although current wake models do not include this information. Finally, the streamwise behavior of eddy viscosity has yet to be quantified in a parametric study spanning multiple inflow conditions, turbine sizes, and misalignment angles."

**Comment 2:** Your chosen method for obtaining an eddy viscosity from a data set is not well motivated and tested. If you use the Boussinesq approximation, then there are 5 equations but you only have one unknown: the scalar eddy viscosity. Hence, the system is over determined. You have chosen to only look at $\overline{u'w'}$ and then neglect the other equations and you motivate this in Lines 129-132: *Eddy viscosity values are obtained at each downstream location from the slope of a*

*least squares linear regression between $S_{13}|_w$ and $\overline{u'w'}|_w$. The streamwise-vertical components of the Reynolds stress and rate of strain tensors are selected as they are of the greatest magnitude in the wake and are responsible for the majority of energy flux into a wind plant (Porté-Agel et al., 2020; Scott et al., 2020).* I agree that the shear-stresses are the main contributors to the wake recovery, however, $S_{13}|_w$ does not have to the largest component, it can also be the lateral Reynolds-stress $\overline{u'v'}|_w$, depending on the inter wind turbine spacing in the wind farm, see for example []. In a recent work of my own [], I showed/visualized that the peaks of the shear stresses indicate how much wake recovery takes places in the lateral and vertical directions. You could look at your own LES and wind tunnel data to see which is dominant one. Alternatively, you could also write the Boussinesq hypothesis in polar coordinates and then you could choose to only use the equation for the shear stress in the radial direction for calculating the eddy viscosity from the data. This also makes sense for your application where the employed engineering wake models uses axisymmetry. If you choose to stick with the current method, you should at least show how large the errors are from the unused equations of the Boussinesq hypothesis, especially the one for the lateral shear stress.

**Response:** Thank you for these suggestions, we have responded to each individually here and in Comment 6:

The regression method for determining eddy viscosity is a direct application of the Boussinesq equation for $i \neq j$. This technique has been successfully demonstrated by Bai et al. [1] and more recently by Rockel et al. [13]. In the present work, this method facilitated identical treatment of both LES and experimental data ensuring conclusions drawn across data sets reflect the underlying phenomenon rather than processing technique.

The initial formulation employed a curvilinear polar coordinate system where we solved for the radial shear stress as you have suggested. However, this approach possesses technical difficulties which lead to unsatisfactory errors when computing eddy viscosity. Because wake center tacking is necessary for mapping a polar coordinate system to the wake of a misaligned turbine, case-specific coordinate fit errors are permuted through subsequent computations. This problem is compounded for tilt cases where the wake is deflected into the ground due to the rapid deformation and dissipation experienced by such wakes.

Prior to selecting the streamwise-vertical components, we computed eddy viscosity for both the streamwise-vertical and streamwise-lateral Boussinesq equations. The streamwise-vertical eddy viscosity was dominant in every case except for $\pm 20°$ yaw where the two equations yielded eddy viscosities of comparable order. The streamwise-lateral eddy viscosity was far lower in magnitude across tilt cases. We acknowledge the cases where the streamwise-lateral components are significant and have have expanded on our reasons for selecting the streamwise-vertical comoponents in response to Comment 6.

**Comment 3:** If you want make general conclusions about the proposed analytical model of the eddy viscosity then I would recommend to investigate aligned single wake wind turbines first using a range of thrust coefficients and turbulence intensities, since these two parameters are the main ones. I am aware that misaligned turbines will typically have lower thrust coefficients, but you will also have a lateral thrust component that changes the wake shape. (Atmospheric stability is also important and could be added in future work.)

**Response:** Thank you for this interesting idea. While such a study is outside the scope of the current manuscript, we agree this suggestion would make for excellent future work. Accounting for the model response to atmospheric conditions and variations in turbine performance is key to

ensuring it performs as expected. We have recommended this suggestion for future studies in the manuscript:

"Further parameterizion to include multiple turbulence intensities, turbine thrust coefficients, and atmospheric stabilities would ensure the proposed model performs across settings."

**Comment 4:** What is a Hybrid wake model? In the introduction you write that a Hybrid wake model has the computational efficiency of an analytic/engineering wake model. To me that sounds like a Hybrid wake model falls into the class of engineering wake models. In my personal opinion, any model that does not solve a momentum equation iteratively (i.e. including pressure gradients) is a low fidelity / engineering wake model.

**Response:** This is an important point for clarification, thank you for highlighting it. In this case hybrid denotes models which solve a parabolic or linearized form of the RANS equations [9, 10, 3]. While these models are inherently a simplified representation of the complete flow, they include physics which are not present in superposition based models. The introductory text now includes:

"Accurate wake modeling is essential for optimizing wind plant layouts and creating effective control strategies [17, 11]. Hybrid wake models balance the accuracy of high fidelity simulations with the computational efficiency of analytic models to facilitate wind plant design studies. Unlike superposition based approaches, hybrid wake models adopt a combined RANS-analytic framework to solve a linearized or parabolic representation of the mass and momentum equations [9, 10, 3]. This allows hybrid wake models to include additional physics beyond the scope of typical engineering wake models without incurring substantial computational costs."

**Comment 5:** Lines 34-36: You write *Alternatively, constant eddy viscosities can be modeled with a scalar function tuned to the turbulent production and dissipation of calibration flow (van der Laan et al., 2015).* Note that we are not modeling a constant eddy-viscosity in the work that you refer to. We actually use a three-dimensional scalar eddy viscosity similar to the standard $k - \varepsilon$ model. The model is then improved by multiplying the eddy viscosity with a scalar function that is dependent on the local flow. Effectively, this scalar function limits the eddy viscosity in regions where the flow is far from its equilibrium, i.e. the near wake. The model can also be interpreted a turbulence length scale limiter, see for example []. In theory, you could use the $k - \varepsilon - f_P$ definition to determine the eddy viscosity, which should lead to better prediction of $\nu_T$ in the near wake, but it could be future work to test other forms of $\nu_T$. (A small note: my is spelled as van der Laan.)

**Response:** Our apologies for both the misunderstanding and misspelling. The revised text for this section may be found in our response to Comment 1.

**Comment 6:** Equations (2), (5)-(9) are not complete and should include a term with the turbulent kinetic energy, $k$:

$$\overline{u_i' u_j'} = \frac{2}{3} k \delta_{ij} - 2\nu_T S_{ij}$$

If you aim to neglect $k$ you should mention this and motivate reader that this makes sense. Later on you calculate the eddy-viscosity from only $\overline{u'w'}$, so I understand where omitting $k$ comes from. You could move the motivation from that part (Section 4.1) to theoretical part (Section 2). In addition, I can think equations (7) and (8) can be removed, because the previous equations already define everything.

**Response:** Thank you for these suggestions, we have updated the theory related to Equation (2) and relocated Section 4.1. We however wish to retain Equations (7) and (8) as they demonstrate the far downstream behavior and eventual fate of $\nu_{T,\mathrm{w}}$ once the wake flow is fully recovered. The updated text now reads:
"The eddy viscosity hypothesis relates turbulent stresses to turbulent kinetic energy and the rate of strain tensor. This relationship is introduced as:

$$\overline{u_i'u_j'} = \frac{2}{3}k\delta_{ij} - 2\nu_T S_{ij},$$

where $\overline{u_i'u_j'}$ is the turbulent stress tensor, $k$ is the turbulent kinetic energy, and $S_{ij}$ is the rate of strain tensor. Eddy viscosity is written as $\nu_T$ and acts as a constant of proportionality. In a wind plant, the streamwise-vertical components of the Reynolds stress are responsible for the majority of energy flux into the plant [12, 15] allowing Eq. () to be described in terms of mean flow components:

$$\overline{u_1'u_3'} = -2\nu_T S_{13},$$

Note, in presence of high veer, Coriolis forces, or nacelle yaw the streamwise-lateral stresses are of similar order. In these instances, we expect comparable eddy viscosity magnitudes could be obtained from the streamwise-lateral components."
We have also recommended a future study focused on the streamwise lateral components:
"Detailing the streamwise-lateral rate of strain and shear stress response to yaw, veer, and Coriolis forces is another potential avenue for improving upon the proposed model."

**Comment 7:** You mention that $S_{ij}|_\mathrm{B}$ is assumed to be constant in the fully developed ABL. However, even for a simple neutral surface layer following a logarithmic law we get that $S_{ij}|_\mathrm{B}$ is far from constant and scales with $1/z$: $S_{ij}|_\mathrm{B} = 1/2u_*/(kz)$. If you aim to only look at one height, e.g. hub height, then I can follow your assumption but this does seem to be case in the remaining part of the article (i.e. Figure 4).

**Response:** Thank you, we assume the boundary layer is fully developed, does not vary in the streamwise direction, and is a function of the wall-normal direction only. The text has been updated to avoid confusion and now reads:
"In a fully developed boundary layer, $S_{ij}|_\mathrm{B}$ is assumed not to vary in the streamwise direction and behave as a function of the wall-normal direction only. Therefore, $\nu_{T,\mathrm{B}}$ is independent of $x$."

**Comment 8:** You mention *Thus our efforts focus on modeling $\nu_{T,w}$ in the range where $S_{ij}|_w > 0$*. However, $S_{ij}|_w$ can also be negative in the wake. (That is why $\overline{u'v'}$ has both positive and negative values in the wake.)

**Response:** Good point, because our model is developed from the isolated wake flow it requires $|S_{ij}|_\mathrm{w}| > 0$ and $|\overline{u'w'}|_\mathrm{w}| > 0$ rather than $S_{ij}|_\mathrm{w} > 0$. The text has been updated to match and for clarity:
"Thus our efforts focus on modeling $\nu_{T,\mathrm{w}}$ in the range where the wake flow exists."

**Comment 9:** Section 3.2, LES setup. It is not clear to me what kind of neutral ABL is simulated. Is this a pressure-driven ABL or is this an ABL including an inversion and Coriolis forces? The first type of ABL should be Reynolds number independent for a fixed ratio of the domain height $L_z$ and roughness length, $z_0$, i.e. $z_0/L_z$ is constant. In that case, the two LES cases

mainly differ in wind turbine model if you have scaled the ABL inflow such $z_H/L_z$ is constant. I expect the thrust coefficient to be main difference between the two LES cases, it would be helpful to report the mean thrust coefficient value for both of them. The thrust force distribution could also play a role in the near wake but is not expected to have a large impact on the far wake; a similar argument can be applied for the ground clearance, $z_H/D$ and other design parameters.

**Response:** Thank you for this suggestion, the description of the LES setup has been expanded include:
"Inflow to the domain was driven by a pressure gradient which was adjusted at each timestep to maintain the desired hub-height velocity [5]. The measured thrust coefficients for each turbine configuration at the specified condition are presented in Tab. 1."

| $C_T$ | Yaw | | | | — | Tilt | | | |
|---|---|---|---|---|---|---|---|---|---|
| $\theta$ | $-20°$ | $-10°$ | $10°$ | $20°$ | $0°$ | $-20°$ | $-10°$ | $10°$ | $20°$ |
| 15 MW | 0.79 | 0.84 | 0.84 | 0.78 | 0.86 | 0.71 | 0.81 | 0.89 | 0.82 |
| 1.5 MW | 0.71 | 0.74 | 0.74 | 0.71 | 0.75 | 0.65 | 0.72 | 0.75 | 0.72 |

Table 1: Time-averaged thrust coefficients for the 15 MW and 1.5 MW LES cases.

**Comment 10:** Line 160: you mention *At extreme downstream distances, $x/D > 20$, the eddy viscosity hypothesis no longer holds as both $S_{13}|_w$ and $\overline{u'w'}|_w$ are near zero.* I do not agree, the eddy viscosity hypothesis holds better for a flow that is in near equilibrium (the far-far wake or the background flow) with respect to the (near) wake. I guess your main point refers to the regression method not being able to calculate an eddy-viscosity with the proposed method. Hence this part needs some rewriting.

**Response:** Thanks for highlighting this issue, you are correct the regression approach is ill-suited for fitting two very small quantities. However, the eddy viscosity of the isolated wake flow far downstream will approach zero since $S_{13}|_w$ and $\overline{u'w'}|_w$ are themselves near zero. The text has been amended to clarify our observations are a consequence of working with near-zero quantities in the very far wake:
"At large downstream distances, $x/D > 20$, the wake flow has dissipated and both $S_{13}|_w$ and $\overline{u'w'}|_w$ are near-zero. As the wake has returned to the background flow, performing a linear regression on wake flow components produces erroneous values. This is not the case for the background flow which is treated separately."

**Comment 11:** Equation (11) and (12): I do not understand where Equation (11) comes from, it is not the Boussinesq hypothesis (without $k$) because you lack the gradients and the minus sign. In addition, you could arrive at Equation (12) in a more simply way by dimensional analysis since we have that the unit of $\nu_T$ is m2/s thus one can write $\nu_T = l_s U_s$. In addition, your chosen $U_s$ is based on the far wake of 1D momentum theory, which I think you should mention.

**Response:** Thank you for this suggestion. We chose to perform a scale analysis on Equations (11) and (12) because this technique retains constants such as the 2 associated with $\nu_T$ which are lost with a dimensional analysis. We have modified the text in this section to better describe the scale analysis approach. We have included additional information on our choice of $U_s$ as well. The updated text now reads:

"$A$ is determined by performing a scale analysis on the eddy viscosity hypothesis in which each component of Eq. (2) is written in terms of their respective units. By selecting a velocity scale $U_s$, a length scale $l_s$, and noting $\partial W / \partial x << 1$, we can write:

$$U_s^2 \sim 2\nu_{T,\mathrm{w}} \left[ \frac{U_s}{l_s} \right],$$

Rearranging to isoalte eddy viscosity yields:

$$\frac{l_s U_s}{2} \sim \nu_{T,\mathrm{w}},$$

The wake velocity scale is selected as $U_s \sim U_B \sqrt{1 - C_T}$ following Bastankhah et al. (2016) where $C_T$ is the turbine thrust coefficient and $U_B$ is the mean inflow velocity at hub height. The length scale is selected as $l_s \sim R$ where $R$ is the rotor radius. Note, radius is selected rather than diameter as both the rate of strain tensor and shear stresses are symmetric about the wake center. Additionally, the chosen velocity scale is derived from $1D$ momentum theory to estimate the mean velocity in the far wake. Substituting the velocity and length scales into into Eq. (12) yields an expression for the eddy viscosity magnitude:"

$$\nu_{T,\mathrm{w}} \sim \frac{R U_\mathrm{B} \sqrt{1 - C_T}}{2}$$

**Comment 12:** Lines 258-259: You conclude to have investigated multiple inflow conditions, but you mainly used one LES ABL inflow as the wind tunnel wake data was not sufficient to capture the entire wake.

**Response:** Thank you, we concur the inflow between experimental and LES cases are similar. We have replaced "multiple inflow conditions" with "in a neutral boundary layer".

**Comment 13:** Conflict of interest: I guess this can be removed as Jens Sørensen is the editor. It should not be a problem if a coauthor is also an editor in the journal, as long as he or she is not the handling editor.

**Response:** Thanks, we now declare no competing interests.

**Comment 14:** I think you author contribution statement is too vague. Please clarify who did what.

**Response:** Thank you, the author contribution statement is more specific and now reads:
"R.S. drafted the manuscript; L.M.T., N.H., and R.B.C. edited; J.B. performed the experimental data collection; R.S., L.M.T., and N.H. performed the LES simulations; R.S., L.M.T., N.H., and R.B.C contributed to the model development; R.S. and L.M.T. implemented the model in FLORIS; L.M.T., N.H., and R.B.C. advised."

**Comment 15:** You sometimes write *LES simulation*. This means that you write the word simulation twice so I would suggest to remove the second word.

**Response:** Fixed, thank you!

**Comment 16:** Equation (1): You forgot to mention that you assume an incompressible flow.

**Response:** Thanks for pointing this out, the first line now reads:
"The Reynolds averaged Navier-Stokes equations for incompressible flow are presented in tensor notation as:"

$$u_j \frac{\partial u_i}{\partial x_j} = -\frac{1}{\rho}\frac{\partial p}{\partial x_i} - \frac{\partial \overline{u_i' u_j'}}{\partial x_j} - f_i \tag{1}$$

**References**

[1]   Kunlun Bai, Charles Meneveau, and Joseph Katz. "Near-wake turbulent flow structure and mixing length downstream of a fractal tree". In: *Boundary-layer meteorology* 143.2 (2012), pp. 285–308.

[2]   Majid Bastankhah et al. "A vortex sheet based analytical model of the curled wake behind yawed wind turbines". In: *Journal of Fluid Mechanics* 933 (2022).

[3]   Majid Bastankhah et al. "Analytical solution for the cumulative wake of wind turbines in wind farms". In: *Journal of Fluid Mechanics* 911 (2021).

[4]   Mads Baungaard et al. "Wind turbine wake simulation with explicit algebraic Reynolds stress modeling". In: *Wind Energy Science* 7.5 (2022), pp. 1975–2002.

[5]   Matthew Churchfield et al. "A large-eddy simulation of wind-plant aerodynamics". In: *50th AIAA aerospace sciences meeting including the new horizons forum and aerospace exposition.* 2012, p. 537.

[6]   M Paul van der Laan et al. "An improved $k - \epsilon$ model applied to a wind turbine wake in atmospheric turbulence". In: *Wind Energy* 18.5 (2015), pp. 889–907.

[7]   Maarten Paul van der Laan, Mads Baungaard, and Mark Kelly. "Brief communication: A clarification of wake recovery mechanisms". In: *Wind Energy Science Discussions* (2022), pp. 1–8.

[8]   MP van der Laan and SJ Andersen. "The turbulence scales of a wind turbine wake: A revisit of extended k-epsilon models". In: *Journal of Physics: Conference Series.* Vol. 1037. 7. IOP Publishing. 2018, p. 072001.

[9]   Luis A Martínez-Tossas et al. "The aerodynamics of the curled wake: a simplified model in view of flow control". In: *Wind Energy Science* 4.1 (2019), pp. 127–138.

[10]  Luis A Martínez-Tossas et al. "The curled wake model: a three-dimensional and extremely fast steady-state wake solver for wind plant flows". In: *Wind Energy Science* 6.2 (2021), pp. 555–570.

[11]  Johan Meyers et al. "Wind farm flow control: prospects and challenges". In: *Wind Energy Science Discussions* (2022), pp. 1–56.

[12]  Fernando Porté-Agel, Majid Bastankhah, and Sina Shamsoddin. "Wind-turbine and wind-farm flows: a review". In: *Boundary-Layer Meteorology* 174.1 (2020), pp. 1–59.

[13]  Stanislav Rockel et al. "Wake to wake interaction of floating wind turbine models in free pitch motion: An eddy viscosity and mixing length approach". In: *Renewable Energy* 85 (2016), pp. 666–676.

[14]  Martin Schmelzer, Richard P Dwight, and Paola Cinnella. "Discovery of algebraic Reynolds-stress models using sparse symbolic regression". In: *Flow, Turbulence and Combustion* 104.2 (2020), pp. 579–603.

[15]  Ryan Scott, Juliaan Bossuyt, and Raúl Bayoán Cal. "Characterizing tilt effects on wind plants". In: *Journal of Renewable and Sustainable Energy* 12.4 (2020), p. 043302.

[16]  Carl R Shapiro, Dennice F Gayme, and Charles Meneveau. "Generation and decay of counter-rotating vortices downstream of yawed wind turbines in the atmospheric boundary layer". In: *Journal of Fluid Mechanics* 903 (2020).

[17]   Paul Veers et al. "Grand Challenges in the Design, Manufacture, and Operation of Future Wind Turbine Systems". In: *Wind Energy Science Discussions* (2022), pp. 1–102.

---

## Author Comment (AC2)

In this paper, the authors have described a new analytical method to determine the eddy viscosity from LES and wind tunnel experiments. The authors incorporated the eddy viscosity model with with an engineering wake model and compared the results with that from LES. Overall, this paper is well-written, proposes a novel idea to determine eddy viscosity and have great potential to improve the accuracy of evaluations of wind farm wake. I really enjoyed reading it. Overall, I recommend publication in Wind Energy Science, but I think some very minor adjustments and comments might further improve this paper.

**Response:** The authors appreciate this review and the constructive comments. The text has been edited to reflect the reviewers comments and for clarity. The reviewers comments are addressed below and further revisions are highlighted in **bold** in the text.

**Comment 1:** Starting Line 12 and multiple instances: Please provide reference and/or define "hybrid wake model".

**Response:** Thank you for this suggestion, we have updated the introduction to contextualize hybrid wake models:
"Accurate wake modeling is essential for optimizing wind plant layouts and creating effective control strategies [7, 4]. Hybrid wake models balance the accuracy of high fidelity simulations with the computational efficiency of analytic models to facilitate wind plant design studies. Unlike superposition based approaches, hybrid wake models adopt a combined RANS-analytic framework to solve a linearized or parabolic representation of the mass and momentum equations [2, 3, 1]. This allows hybrid wake models to include additional physics beyond the scope of typical engineering wake models without incurring substantial computational costs."

**Comment 2:** Line 46 −− 48: LES simulations − > LES

**Response:** Fixed, thank you!

**Comment 3:** Equation 2: Please add when $i \neq j$. When $i = j$, the full form of Boussinesq approximation has a term of $2/3k$.

**Response:** Thank you, we have updated the theory related to Equation (2) and relocated Section 4.1. The updated text now reads:
"The eddy viscosity hypothesis relates turbulent stresses to turbulent kinetic energy and the rate of strain tensor. This relationship is introduced as:

$$\overline{u_i' u_j'} = \frac{2}{3} k \delta_{ij} - 2\nu_T S_{ij},$$

where $\overline{u_i' u_j'}$ is the turbulent stress tensor, $k$ is the turbulent kinetic energy, and $S_{ij}$ is the rate of strain tensor. Eddy viscosity is written as $\nu_T$ and acts as a constant of proportionality. In a wind plant, the streamwise-vertical components of the Reynolds stress are responsible for the majority of energy flux into the plant [5, 6] allowing Eq. () to be described in terms of mean flow components:

$$\overline{u_1' u_3'} = -2\nu_T S_{13},$$

Note, in presence of high veer, Coriolis forces, or nacelle yaw the streamwise-lateral stresses are of similar order. In these instances, we expect comparable eddy viscosity magnitudes could be

obtained from the streamwise-lateral components."

**Comment 4:** Line 147: How did you normalize eddy viscosity? Please add it in the main text.

**Response:** Thank you for pointing this out. In Figure 5, $\nu_{T,\mathrm{w}}^{\star}$ is relative to the maximum for that case while in the remainder of the manuscript $\nu_{T,\mathrm{w}}^{\star}$ is given by:

$$\nu_{T,\mathrm{w}}^{\star} = \nu_{T,\mathrm{w}}(x)/A\left[0.01 + \frac{x}{\sigma^2}e^{-x^2/2\sigma^2}\right]$$

where $A = RU_{\mathrm{B}}\sqrt{1-C_T}/2$ and $\sigma = 5.5$. We have added the following near line 147 for clarity: "Here, $\nu_{T,\mathrm{w}}^{\star}$ is normalized relative to the maximum value for each case to facilitate consistent comparisons across cases."

**Comment 5:** Overall comment for Section 3.2, 4.1 and 4.2: It may be helpful if more runs for LES are performed with different thrust coefficients for the wind turbine the coefficient $\sigma$ in Eq 14 is obtained in a statistical way.

**Response:** Thank you, we agree parameterizing the model response to various inflow conditions and turbine operations is a valuable next step. We have addressed this suggestion alongside Comment 6.

**Comment 6:** Overall comment for Section 5: The proposed eddy viscosity model comes from the neutral ABL. Just curious whether the authors are planning to verify whether the proposed model still works under different atmospheric stability conditions.

**Response:** Thank you for this and the prior suggestion. While such studies are outside the scope of the present work we have suggested both a future work:
"Further parameterizion to include multiple turbulence intensities, turbine thrust coefficients, and atmospheric stabilities would ensure the proposed model performs across settings. Additionally, future work can resolve the discrepancies reported for nacelle misalignment. Describing surface interactions in terms of turbine operating parameters and roughness height is one promising avenue for further refinement. Detailing the streamwise-lateral rate of strain and shear stress response to yaw, veer, and Coriolis forces is another potential avenue for improving upon the proposed model. We anticipate future developments in this area will lead to improved predictions of wind plant performance and enable the design of more efficient wind plants."

**References**

[1]  Majid Bastankhah et al. "Analytical solution for the cumulative wake of wind turbines in wind farms". In: *Journal of Fluid Mechanics* 911 (2021).

[2]  Luis A Martínez-Tossas et al. "The aerodynamics of the curled wake: a simplified model in view of flow control". In: *Wind Energy Science* 4.1 (2019), pp. 127–138.

[3]  Luis A Martínez-Tossas et al. "The curled wake model: a three-dimensional and extremely fast steady-state wake solver for wind plant flows". In: *Wind Energy Science* 6.2 (2021), pp. 555–570.

[4]  Johan Meyers et al. "Wind farm flow control: prospects and challenges". In: *Wind Energy Science Discussions* (2022), pp. 1–56.

[5] Fernando Porté-Agel, Majid Bastankhah, and Sina Shamsoddin. "Wind-turbine and wind-farm flows: a review". In: *Boundary-Layer Meteorology* 174.1 (2020), pp. 1–59.

[6] Ryan Scott, Juliaan Bossuyt, and Raúl Bayoán Cal. "Characterizing tilt effects on wind plants". In: *Journal of Renewable and Sustainable Energy* 12.4 (2020), p. 043302.

[7] Paul Veers et al. "Grand Challenges in the Design, Manufacture, and Operation of Future Wind Turbine Systems". In: *Wind Energy Science Discussions* (2022), pp. 1–102.

---

## Referee Report (RR1)

**Review of *Evolution of Eddy Viscosity in the Wake of a Wind Turbine, R1* by Ryan Scott et al.**

Reviewer: M. Paul van der Laan, DTU Wind Energy

February 7, 2023

I would like to thank the authors for their answers and revised article. The authors have mostly responded correctly to my comments and I only have a few remaining comments regarding the revised text:

**Main comments**

1. The response to Comments 1 and 5 is not correct. Page 2, Line 38: You have added *Alternatively, constant eddy viscosities can be modeled with a scalar function tuned to the turbulent production and dissipation of calibration flow (van der Laan et al., 2015*. This work does not consider a constant eddy viscosity as your text suggests. Instead, the standard $k$-$\varepsilon$ model definition of the eddy viscosity, $\nu_T = C_\mu k^2/\varepsilon$, is multiplied by a variable scalar function, $f_P$: $\nu_T = f_P C_\mu k^2/\varepsilon$. This $f_P$ function acts as a turbulence length scale limiter in the near wake; the resulting eddy viscosity is a three dimensional scalar variable.

2. The LES case description is more clear now. The only two things I am still missing is the ambient turbulence intensity (based on TKE) at hub height for each turbine case and the value of the applied roughness length at the ground.

**Minor comments**

1. Page 8 , Line 177: The revised derivation of the analytic eddy viscosity is more clear now, but it still contains a wrong reference to an equation. You write: *A is determined by performing a scale analysis on the eddy viscosity hypothesis in which each component of Eq. (2) is written in terms of their respective units.* I think Eq. (2) should be Eq. (3).

**References**

---

## Author Response (AR2)

I would like to thank the authors for their answers and revised article. The authors have mostly responded correctly to my comments and I only have a few remaining comments regarding the revised text:

**Response:** The authors appreciate your follow through and continued work to improve our manuscript. The text has been edited to reflect the reviewers comments and for clarity. The reviewers comments are addressed below and further revisions are highlighted in **bold** in the text.

**Comment 1:** The response to Comments 1 and 5 is not correct. Page 2, Line 38: You have added "Alternatively, constant eddy viscosities can be modeled with a scalar function tuned to the turbulent production and dissipation of calibration flow (van der Laan et al., 2015)". This work does not consider a constant eddy viscosity as your text suggests. Instead, the standard $k - \varepsilon$ model definition of the eddy viscosity, $\nu_T = C_\mu k^2/\varepsilon$, is multiplied by a variable scalar function, $f_P : \nu_T = f_P C_\mu k^2/\varepsilon$. This $f_P$ function acts as a turbulence length scale limiter in the near wake; the resulting eddy viscosity is a three dimensional scalar variable.

**Response:** Thank you for bringing this up, our interpretation was based on selecting a global Rotta constant to tune $f_P$. However, we recognize this neglects the local flow dependence captured by $\sigma/\tilde{\sigma}$. We have revised the text accordingly:
"Alternatively, a three-dimensional eddy viscosity can be modeled with a scalar function tuned to the turbulent production and dissipation of calibration flow (van der Laan et al., 2015; van der Laan and Andersen, 2018). This scalar functions acts as a turbulence length scale limiter which allows the model to represent localized behavior in the near wake and at the wake edges as well as improving velocity deficit estimation."

**Comment 2:** The LES case description is more clear now. The only two things I am still missing is the ambient turbulence intensity (based on TKE) at hub height for each turbine case and the value of the applied roughness length at the ground.

**Response:** Thank you for these suggestions, we have specified surface roughness and TKE based turbulence intensity in the text:
"A neutral atmospheric boundary layer inflow was generated with a $20,000$ second precursor simulation on each base domain with a hub-height inflow velocity of $8$ ms$^{-1}$ and a surface roughness of $0.15$ m. Hub-height turbulence intensities were computed from turbulent kinetic energy and averaged from $x/D = -0.25$ to $x/D = -1.25$. The mean turbulence intensity for the 15 MW cases was $6.2 \pm 0.3\%$ and the mean turbulence intensity for the 1.5 MW cases was $8.4 \pm 0.2\%$."

**Comment 3:** Page 8 , Line 177: The revised derivation of the analytic eddy viscosity is more clear now, but it still contains a wrong reference to an equation. You write: "$A$ is determined by performing a scale analysis on the eddy viscosity hypothesis in which each component of Eq. (2) is written in terms of their respective units". I think Eq. (2) should be Eq. (3).

**Response** You are correct, thank you.